# Waffle Method: A general and flexible approach for improving throughput in FIB-milling

Kotaro Kelley[1,6], Ashleigh M. Raczkowski [1,4,5], Oleg Klykov[1,2,5], Pattana Jaroenlak[3,5], Daija Bobe [1,5], Mykhailo Kopylov[1], Edward T. Eng [1], Gira Bhabha[3], Clinton S. Potter[1,2], Bridget Carragher [1,3 ✉] & Alex J. Noble [1 ✉]

Cryo-FIB/SEM combined with cryo-ET has emerged from within the field of cryo-EM as the method for obtaining the highest resolution structural information of complex biological samples in-situ in native and non-native environments. However, challenges remain in conventional cryo-FIB/SEM workflows, including milling thick specimens with vitrification issues, specimens with preferred orientation, low-throughput when milling small and/or low concentration specimens, and specimens that distribute poorly across grid squares. Here we present a general approach called the 'Waffle Method' which leverages high-pressure freezing to address these challenges. We illustrate the mitigation of these challenges by applying the Waffle Method and cryo-ET to reveal the macrostructure of the polar tube in microsporidian spores in multiple complementary orientations, which was previously not possible due to preferred orientation. We demonstrate the broadness of the Waffle Method by applying it to three additional cellular samples and a single particle sample using a variety of cryo-FIB-milling hardware, with manual and automated approaches. We also present a unique and critical stress-relief gap designed specifically for waffled lamellae. We propose the Waffle Method as a way to achieve many advantages of cryo-liftout on the specimen grid while avoiding the long, challenging, and technically-demanding process required for cryo-liftout.

[1] National Center for In-situ Tomographic Ultramicroscopy, Simons Electron Microscopy Center, New York Structural Biology Center, New York, NY, USA. [2] Department of Biochemistry and Molecular Biophysics, Columbia University, New York, NY, USA. [3] Skirball Institute of Biomolecular Medicine and Department of Cell Biology, New York University School of Medicine, New York, NY, USA. [4] Present address: Cryo-Electron Microscopy Lab, Life Sciences Institute, University of Michigan, Ann Arbor, MI, USA. [5] These authors contributed equally: Ashleigh M. Raczkowski, Oleg Klykov, Pattana Jaroenlak, Daija Bobe. [6] Deceased: Kotaro Kelley. ✉email: bcarr@nysbc.org; anoble@nysbc.org

Cryo-focused ion beam milling scanning electron microscopy (cryo-FIB/SEM)[1] of intact cells plunge frozen onto EM grids (conventional cryo-FIB/SEM) followed by cryo-electron tomography (cryo-ET) is developing as a fruitful sample preparation and analysis method for structural studies of cells thinner than about 10 μm. In some instances, cryo-FIB/SEM followed by cryo-ET (cryo-FIB/SEM-ET) and sub-tomogram processing has produced in-situ 3D structures of molecular complexes at resolutions on the order of 1 nm[2–5]. Methods for promoting cell adhesion onto grids have been developed[6–8]. However, the broad application of conventional cryo-FIB/SEM with reasonable throughput is often challenging for thick specimens with vitrification issues and specimens with dimensions smaller than about 2 μm which adopt a preferred orientation on the grid.

Three previous approaches have been used to address issues of specimens thicker than the ~50 μm limit (the typical depth of focus limit for gallium FIB beams)[9] prior to FIB-milling lamellae for imaging in a transmission electron microscope (TEM). Cryo-ultramicrotomy has been used to thin vitrified tissues from hundreds of microns thick to 20–30 μm prior to conventional cryo-FIB wedge milling[10]. Recently, cryo-microtomy and cryo-FIB-mill trenching of vitrified *C. elegans* in-situ followed by micromanipulator cryo-liftout and further FIB-milling on custom EM grids has been demonstrated[9,11]. Yet few labs possess both the equipment and expertise to carry out such studies. An alternative method for obtaining three-dimensional information from native specimens in-situ is automated cryo-slice-and-view (cryo-ASV)[11]. Cryo-ASV is performed by using a high-pressure freezer (HPF) to vitrify a block of specimen, which is then inserted into a cryo-FIB/SEM for sample manipulation and imaging. The bulk specimen is thinned down until an area of interest is identified and the surface is then iteratively imaged and milled, allowing for the whole volume to be imaged at resolutions of several tens of nanometers in (x, y, z), a resolution which is often a restrictive limit for molecular studies[11].

Here we present a general method, called the 'Waffle Method', for cryo-FIB-milling lamella of a broader range of specimens in-situ compared to conventional cryo-FIB/SEM. The Waffle Method has the following advantages: (1) there are far fewer restrictions on the size of the specimens that can be reliably milled (from nanometers to tens of microns); (2) preferred orientation issues for specimens within this size range are eliminated; and (3) several very large (between 10 × 10 μm and 70 × 25 μm in x,y) and specimen-dense lamellae may be prepared regardless of the specimen size, which substantially increases throughput. Moreover, the Waffle Method inherently solves vitrification issues of all sample types through the use of an HPF, does not require additional FIB/SEM hardware for preparation, and may be used to FIB-mill specimens with thicknesses up to about 50 μm. In developing the Waffle Method, a modified form of gap milling for lamellae stress-relief called notch milling proved to be critical. We describe notch milling and its potential mechanism herein.

To illustrate the advantages of the Waffle Method, we present a comparison of the Waffle Method and conventional cryo-FIB/SEM on native, intact microsporidian spores. Microsporidia are obligate, unicellular, spore-forming parasites that enclose a ~100 μm long polar tube used to infect target host cells[12,13]. The object of interest within the dormant microsporidian spores - the polar tube - is in a fixed orientation relative to the spore[13] and the spores exhibit preferred orientation on the grid, constituting an intractable orientation issue for conventional cryo-FIB/SEM. We show that the Waffle Method solves the preferred orientation issue of the polar tube within the spores while retaining nanometer-level features. Moreover, the Waffle Method allows

for more efficient and higher-throughput FIB-milling of these spores compared to conventional cryo-FIB/SEM due to significantly larger lamella densely packed with cells, which is a generalizable advantage for all specimens of comparable size. To illustrate the broad applicability of the Waffle Method, we apply it to three additional cell types - yeast *S. cerevisiae* cells, *E. coli* cells, and HEK 293 S cells - and use a variety of cryo-FIB/SEM hardware, including the Aquilos 2 combined with automated lamellae milling and manual milling on the Helios with two different cryo-stages.

The Waffle Method may also have the following additional advantages that are currently being explored and are discussed: (1) thin complexes in-vitro with lengths on the order of micrometers to millimeters may be studied structurally from all directions, such as filaments and microtubules; (2) proteins in bulk solution may be studied by single particle cryo-electron microscopy (cryo-EM) or cryo-ET, which may be the only available method amenable to samples that do not behave in the thin ice of conventional cryo-EM grids due to air-water interface issues causing denaturation, aggregation, preferred orientation, etc;[14–16]. (3) very long and skinny cells may be studied more completely; (4) cells may be studied that are difficult to mill conventionally due to suboptimal affinity to grids, suboptimal concentration in grid squares, wicking/hydration issues on grids, etc.; (5) tissues and multi-cellular organisms may be prepared as an alternative to the cryo-FIB/SEM liftout method.

The term 'waffle' quickly caught on during development because a high-pressure frozen sample on a grid with grid bar lines milled onto the top resembles a waffle in an SEM. The name also helps explain the Waffle Method by simply referring to a breakfast waffle with syrup filling in the squares (ie. sample filling squares on an EM grid).

## Results

We first describe the overall workflow for the Waffle Method, which is general and agnostic to specific hardware. Subsequent sections describe workflows and results that were waffled using specific specimens and instruments.

**Overall workflow.** Figure 1 and Supplementary Movie 1 show the overall, general Waffle Method workflow. Workflows for each specific sample are detailed in the Methods. Before making the waffle, the hardware needs to be prepared (Step 0): (a) planchette hats should be polished - first with varying grit sandpaper (Supplementary Fig. 1a–e), then with metal polish - to reduce lamellae curtaining, as we substantiate herein; (b) 1-hexadecene is applied and incubated on the planchette hats (Supplementary Fig. 1f) and the optional spacer ring; (c) the EM grid is rigidified by applying ~25 nm layer of carbon to the non-grid bar side of the grid; (d) ethanol is used to clean the HPF tip; and (e) the EM grid is glow discharged/plasma cleaned.

Waffle making (Step 1) is performed as follows: (f) the sample and the planchette hats are assembled into a waffle form with the sample placed on the grid bar side of the EM grid as shown in Supplementary Fig. 2 and Supplementary Movies 1 and 2; (g) the sample is high-pressure frozen; and (h) the planchette hats are removed and the grid is carefully clipped into an autogrid, followed by an optional sputtering of a conductive platinum coat. At this point the waffle grid may optionally be screened in a cryo-FLM.

The waffle grid is then transferred to a flat cryo-FIB/SEM grid holder (or tilted grid holder if near-orthogonal milling to the grid is possible) and then to the cryo-FIB/SEM chamber to mill the waffle (Step 2) as follows: (i) several microns of platinum are deposited using a Gas Injection System (GIS); (j) grid bar lines are

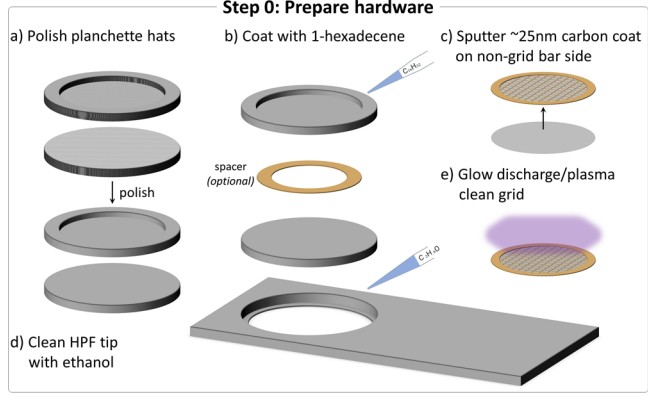

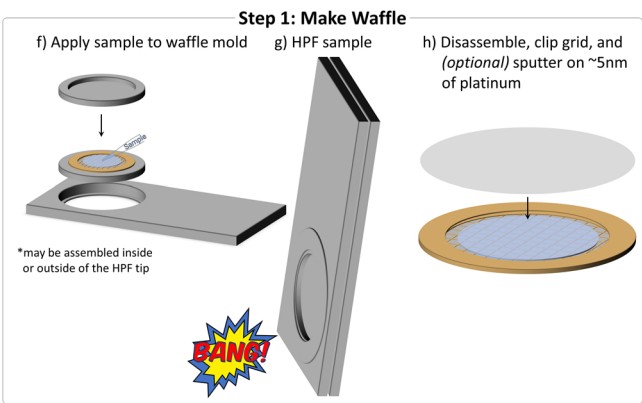

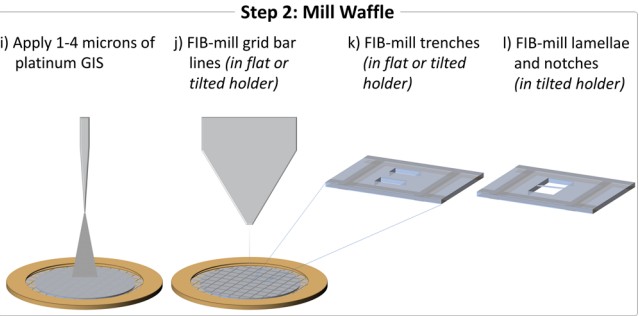

**Fig. 1 Schematic of the general workflow for creating a waffle grid.** Supplementary Movie 1 shows an animation of all steps. In Step 0, the waffle is prepared: **a** The planchette hats are polished to reduce lamellae curtaining, **b** The planchette hats and optional spacer are coated with 1-hexadecene to make easier to separate after high-pressure freezing, **c** ~25 nm of carbon is sputtered onto the non-grid bar side of the EM grid to rigidify it, **d** The high-pressure freezer (HPF) tip is cleaned with ethanol just prior to use, and **e** the EM grid is glow discharged/plasma cleaned just prior to sample application. In Step 1, the waffle is made: **f** The sample is applied to the EM grid with optional spacer on top of a planchette hat, **g** The waffle is vitrified with the HPF, **h** The HPF assembly is taken apart, the grid is carefully clipped, and ~5 nm of conductive platinum is sputtered onto the waffle. The researcher may want to screen their waffle grid in a cryo-FLM at this step. In Step 2, the waffle is milled inside of a cryo-FIB/SEM: **i** 1–4 μm of platinum Gas Injection System (GIS) is applied, (**j**) Grid bar lines are drawn onto the top of the waffled sample to expedite location identification during milling, **k** Two trenches per lamella are milled as perpendicular to the grid as possible and separated by tens of microns, **l** The grid is tilted to a shallow angle and all lamellae on the grid are milled in parallel while making sure the leading edge of the FIB beam always intersects platinum GIS first to reduce curtaining. It is recommended that a notch be milled into one side of each lamellae during the course of milling, as shown in Fig. 2. It is recommended to tilt the stage ±1° while milling the lamellae from 3 μm thick to 1.5 μm thick and ±0.5° while milling the lamellae from 1.5 μm thick to 0.5 μm thick prior to fine milling and polishing at the desired angle. Supplementary Figs. 3 and 4 show waffle milling FIB/SEM images. Supplementary Movie 2 shows most of Step 1 being performed.

Parallel milling reduces contamination, redeposition, and condensation build-up across all lamellae. Supplementary Fig. 3 shows milling details from trench milling to polishing. Supplementary Fig. 4 shows additional images during waffle milling. The finished lamellae are then sputtered with several nanometers of conductive platinum in an inert atmosphere. The waffle grid with lamellae is then transferred to a TEM for cryo-ET collection followed by 3D analysis. Supplementary Table 1 lists the required and recommended hardware used for the Waffle Method.

**Waffle Method applied to small specimens with preferred orientation issues.** To illustrate several of the advantages of the Waffle Method, we applied the method to dormant microsporidian *Encephalitozoon hellem* (*E. hellem*) spores. *E. hellem* spores are about 1.5 μm in diameter and 2.5–3.5 μm in length. Inside each unactivated microsporidian spore is a long, coiled polar tube with a diameter of ~130 nm, and several other organelles. Due to their shape, microsporidia on conventionally-prepared plunge-frozen EM grids almost exclusively lie on the grid with their long (anterior-posterior) axis parallel to the plane of the grid (Fig. 3a). The polar tube, the most prominent organelle inside the microsporidian spore, is predominantly wound at a tilt around the long axis of the spore on the inside edge of the spore wall[13] (Fig. 3e, schematic diagram). This polar tube orientation in the microsporidia and preferred orientation of the microsporidia on the grid, as shown in Fig. 3a, b, e, results in intractable preferred orientation of the polar tube in lamellae after conventional cryo-FIB/SEM milling. Specifically, milling individual microsporidian spores inevitably provides only axial views of the polar tube. Moreover, the throughput and accuracy of milling individual or small groups of microsporidia is low (Fig. 3a, b), and the resulting lamellae are challenging for cryo-ET data collection due to the very small lamella and the location of the polar tube near the spore wall.

We applied the Waffle Method to microsporidia (Fig. 3c–g) in order to visualize the polar tube in dormant spores, to solve the preferred orientation issue, to reliably mill these small cells, to

drawn for convenience; (k) two trenches for each subsequent lamella are milled (>1 nA at an angle of >45° to the plane of the grid) leaving about 30 μm of sample left in the slab between; and (l) the grid is transferred to a tilted cryo-FIB/SEM holder (if not already in one) to mill the lamellae. To protect the resulting lamellae, subsequent milling should always be performed with the ion beam intersecting a part of the slab with a leading edge of platinum GIS. At a shallow angle (~25° from the grid plane), the bottom of the slab is removed (>1 nA) so that a double lamella is less likely, and then verified with an SEM image. At more shallow angles (~20° or less from the grid plane), the slab is thinned down stepwise to about 3 μm thick while reducing the ion beam current. With a similar current (~0.3 nA), a notch with a separation width of about 200 nm is milled completely through one side of the lamella to provide stress-relief (Fig. 2). Current is reduced and the lamellae is milled down to about 1.5 μm thick while tilting the stage ±1° with a tab left beside the notch. Current is reduced further while milling the lamella down to about 500 nm thick while tilting the stage ±0.5°. Tilting the stage small amounts while milling may help keep the lamellae thickness uniform[17]. The lamella is then polished down to the final desired thickness. Each stage in milling should be performed in parallel across each lamella after the lamellae are thinner than several microns.

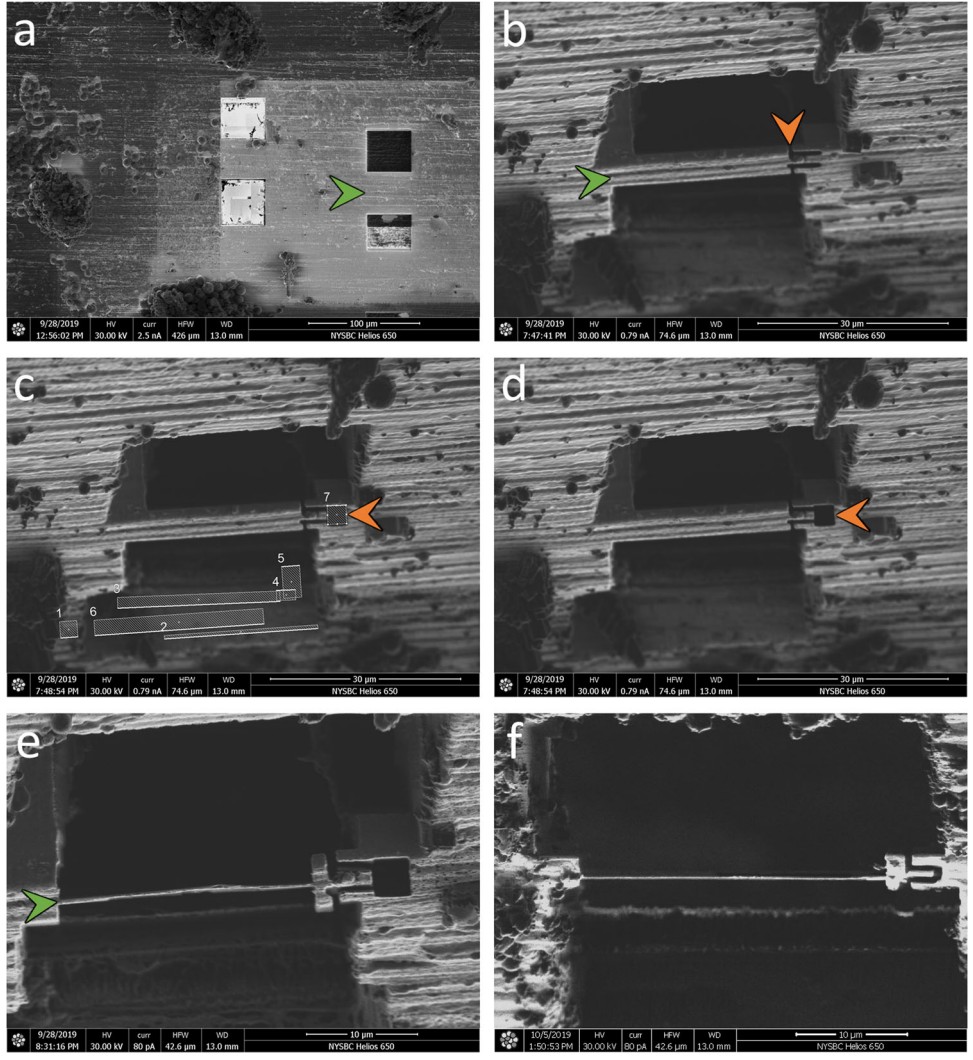

**Fig. 2 FIB/SEM images of the notch milling workflow.** After trench-milling **a** (green arrow), the resulting slab is coarsely milled down to <10 µm thick **b** (green arrow). An initial, incomplete notch design is milled into one side of the slab, with a segment still connected **b** (orange arrow). This connection is then broken by milling, revealing a tab within a notch **c**, **d** (orange arrow). The lamella is then milled and polished as usual **e** (green arrow). A second example of a completed lamella with a notch mill is presented in **f**. To allow for freedom of movement (see Supplementary Fig. 8 and Supplementary Movie 9), each notch mill should separate the tab from the slab by 200+ nm. with similar results.

increase their concentration in lamellae, and to improve the throughput of cryo-FIB/SEM preparation and of cryo-ET collection while retaining nanometer-level features. The following specific workflow was used, which is depicted and shown in Fig. 1 and Supplementary Movies 1 & 2. Full details of the Waffle Method workflow used here are presented in the Methods.

**Prepare hardware and waffle the sample.** To reduce the amount of curtaining that occurs during cryo-FIB-milling, the topography of the top of the waffle should be as uniformly smooth as possible above areas that will be milled (Fig. 3c, Supplementary Fig. 1a–e). Deposited layers of conductive platinum and platinum GIS increase the thickness of the waffle, but do not change the topography substantially. To obtain a smooth top waffle surface, we apply metal polish paste to each solid brass planchette hat using a Kimwipe and polish for several minutes per side until all visible features disappear.

To rigidify the grid, about 25 nm of carbon is sputtered onto the non-grid bar side of the grid. The survivability of grid squares during grid handling and high-pressure freezing is substantially increased if the grid has an additional layer of sputtered carbon.

This is due to several steps in the Waffle Method imparting considerable forces onto the EM grid: The HPF process delivers a substantial impulse to the grid; the resulting waffle often has non-uniform coverage radially, causing differential stress and strain across the remaining open areas of the grid; and several transfer steps increase the risk of grid damage (waffle assembly, waffle disassembly, grid clipping, cryo-FIB/SEM holder loading/unloading, optional cryo-FLM loading/unloading, cryo-TEM loading).

The planchette hats and optional spacer ring are coated with 1-hexadecene to allow for easier disassembly of the waffle after freezing[18,19]. The HPF tip is cleaned with ethanol to remove any contaminants. The waffle grid is assembled in the HPF tip as shown in Supplementary Fig. 2 and Supplementary Movie 2: Several microliters of sample are applied to the grid bar side of the EM grid in the bottom planchette hat, the top planchette hat is placed on, the HPF tip is closed, and excess sample is wicked away with a Kimwipe while being careful not to wick away sample from the grid. The planchette-waffle grid assembly is then quickly transferred to the HPF where it is high-pressure frozen. The assembly is then removed from the HPF tip in LN2 before the planchette hats are disassembled to release the waffled grid.

## Conventional cryo-FIB/SEM

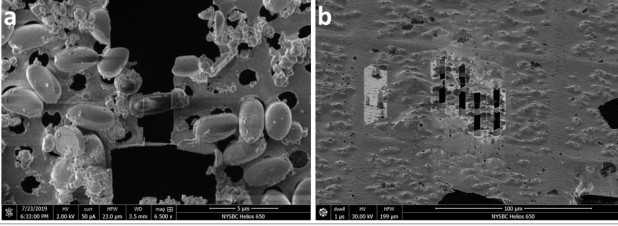

## Waffle Method

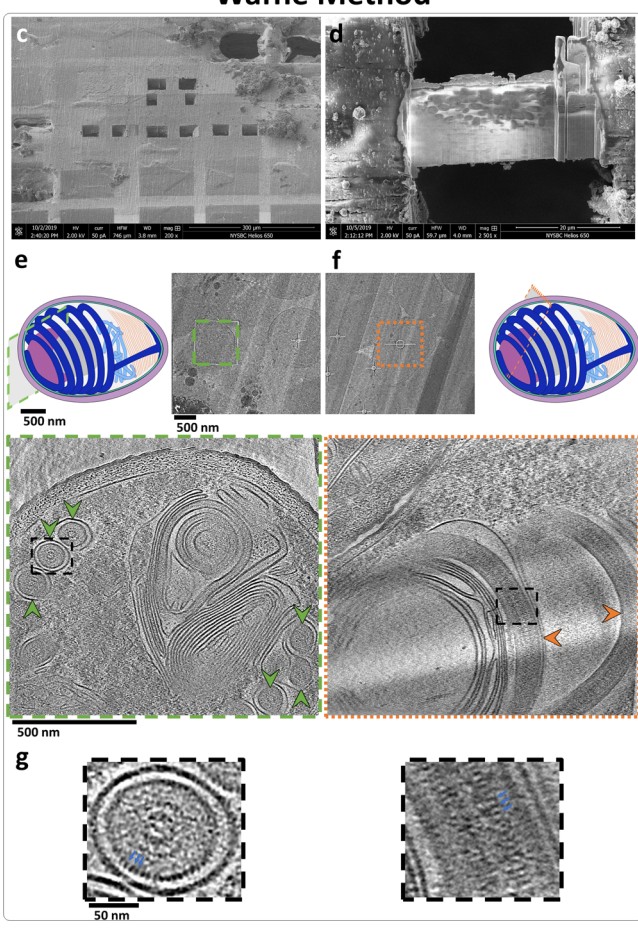

**Fig. 3 Example of how the Waffle Method solves low concentration, low-throughput, and preferred orientation problems of conventional cryo-FIB/SEM of microsporidian spores. a**, **b** Small cells milled by conventional cryo-FIB/SEM where the samples were back-blotted and plunge frozen. The cells are individually-milled due to low concentration, leading to low-throughput. **a** SEM image of an individually-milled cell (~1.5 × 3 μm). **b** FIB image of several individually-milled cells. **c**–**g** Small cells prepared using the Waffle Method. **c** SEM image of a waffle with several trenches prepared. **d** SEM image of a completed waffle lamella (~30 × 20 μm) with a notch mill showing several orientations of the spores. **e** A low-mag TEM image of a waffled microsporidian spore lamella alongside a schematic diagram of a spore with the high-mag cryo-ET collection area approximated (green outlined cross-section). Below is a slice-through of the high-mag tomogram with arrows showing axial views of the polar tube in the spore (green arrows). The polar tube in the schematic diagram is colored dark blue, exhibiting a fixed orientation relative to the major axis of the spore. **f** A low-mag TEM image of a waffled microsporidian spore lamella alongside a schematic diagram of a spore with the high-mag cryo-ET collection area approximated (orange outlined cross-section). The spore cross-section is roughly orthogonal to the spore in **e**, as the diagrams show. Below is a slice-through of the high-mag tomogram with arrows showing side views of the polar tube in the spore (orange arrows). **g** The dashed black line insets in **e** and **f** magnified by 4x highlighting the ~2.5 nm features on the second cylindrical layer (blue arrows). Tomogram slice-through movies are shown in Supplementary Movie 3. **a**, **c**–**g** show *E. hellem* microsporidian spores while (**b**) shows *A. algerae* microsporidian spores. *n* > 25 independent cells observed in various orientations in tomograms.

The waffled grid is then transferred to the cryo-FIB/SEM prep chamber to sputter several nanometers of conductive platinum onto the grid, then to the cryo-FIB/SEM main chamber where several microns of platinum GIS is deposited onto the grid. Two trenches (several tens of microns in dimensions, separated by tens of microns) per area of interest are milled at angles as close to perpendicular to the grid plane as possible in order to 1) reduce the chances of double lamellae, and 2) reduce trench milling time (Fig. 3c). Partway through coarse milling, a notch is milled into one side of each lamellae (Fig. 2 depicts the workflow). The slabs are then milled in parallel at an angle of 17° down to about 200 nm thick (Fig. 3d shows a finished lamella). Several nanometers of conductive platinum are then sputtered onto the grid and lamellae before transferring the grid to a TEM for cryo-ET collection. Specific details of the milling steps are provided in the Methods.

**Analysis of waffled microsporidian spore sample**. Figure 3d shows a waffled microsporidian spore lamella in the SEM. The

lamella is about 30 × 20 μm in size with a notch milled into the right-hand side. Several microns of platinum are still present on the top of the notch while a micron or less are remaining on top of the finished lamella, illustrating the importance of coating the waffle with a sufficient amount of platinum to achieve a sufficiently smooth waffle. Creating a smooth waffle and applying a sufficient amount of platinum GIS is required for milling thin lamellae and minimizing curtaining (Supplementary Fig. 5). In contrast to conventional plunge freezing and cryo-FIB/SEM (Fig. 3a, b), the waffled lamella contains dozens of microsporidia in many orientations and at high concentration, which increases throughput for both cryo-FIB/SEM and cryo-ET (Fig. 3d).

Figure 3d–g depicts how waffled lamellae of microsporidia solves the preferred orientation problem inherent with the specimen and the object of interest in the spores, the polar tube. The microsporidian spores in the waffled lamellae are present in several orientations, allowing for the full 3D structure of the polar tube to be visualized. The low-mag images in Fig. 3e, f show TEM images of the spores, including an oblique-view in Fig. 3f which is not present in conventionally prepared lamellae (Fig. 3a). The schematic diagrams of the spores in Fig. 3e, f depict cross-sections of approximate locations where tomograms were collected. The bottom images in Fig. 3e, f show slice-throughs of 3D tomograms collected at these two locations. Roughly orthogonal orientations of the polar tubes are present in these tomograms (Supplementary Movie 3), uniquely allowing for nanometer-level features of the polar tube to be visualized. Figure 3g shows 4x magnified views of the square dashed black line inset in the bottom images of Fig. 3e, f. These magnified views show an axial view and a side view of polar tubes with bumps that are about 2.5 nm in dimension clearly visible on the second cylindrical layer of the tube.

**Waffle Method applied to three additional cellular specimens**. To illustrate the broad applicability of the Waffle Method, both in

sample type and in workflow, we applied it to three additional cellular specimens: (1) yeast *Saccharomyces cerevisiae* (*S. cerevisiae*) cells, (2) *E. coli* BL21 (DE3) cells infected with *Leviviridae* PP7-PP7 virus capsid proteins, and (3) HEK 293 S GnTI⁻ suspension cells. Each specimen was prepared, waffled, and FIB-milled as described in the Methods.

Each of the four cellular samples presented in this work were propagated, cultured, expressed, or harvested, then centrifuged and resuspended in solution prior to waffle making. Waffle grids were prepared as shown in Fig. 1 and Supplementary Fig. 2.

The microsporidian spore cells in the previous section and the yeast cells were each milled manually on an FEI Helios NanoLab 650. The spore cells were prepared using a Quorum PP3000T prep chamber and cryo-stage while the yeast cells were prepared using a Leica EM VCT500 cryo-stage. The spore cells were first trench milled orthogonally in a flat holder while the yeast cells were trench milled at a 48° angle in a tilted holder, then excess bulk material below the slabs was milled away at shallower angles prior to lamellae thinning.

The *E. coli* cells and HEK 293 S cells were each milled on a TFS Aquilos 2 using AutoTEM to automate the lamellae thinning process after manual trench milling. First, trench milling for both cells was performed in a tilted holder orthogonally to the grid by rotating the stage, then notches were milled close to the lamellae milling angle, followed by automated overnight lamellae milling. Supplementary Movie 4 shows a timelapse of AutoTEM coarse-to-fine milling.

The manually-milled yeast lamella represents a successful attempt to create the largest waffle lamella and resulted in a lamella of size 70 μm × 25 μm in length and width, and 250–500 nm thick (Supplementary Fig. 6). Figure 4a shows a tomogram slice-through from the lamella and Supplementary Movie 5 shows two representative tomograms.

The resulting AutoTEM-automated lamellae for the *E. coli* and HEK 293 S cells were each about 12 μm × 12 μm in length and width, and 200–250 nm thick. Supplementary Movie 6 shows two representative tomograms from the *E. coli* waffle lamellae. The *E. coli* cells expressed *Leviviridae* PP7 virus capsid proteins, which can be seen assembled as virus capsids inside of the bottom-right *E. coli* cell in Fig. 4b and in the left tomogram in Supplementary Movie 6 (highlighted in red-orange). Moreover, the Waffle Method may allow for neighboring cell-cell membranes (highlighted in green) to more easily be studied. Figure 4c and Supplementary Movie 7 shows a tomogram from the HEK 293 S waffle lamellae. The cell membranes show several dozen receptors of interest (several highlighted in blue). The Waffle Method applied to *E. coli* and HEK 293 S cells illustrate the advantage of milling dozens of cells per lamellae to increase the likelihood of locating low-density regions of interest in the resulting cryo-ET tomograms, compared to conventional cryo-FIB-milling. A low-magnification tomogram of the *E. coli* sample (Supplementary Movie 8) shows that sections of the lamella contain complete outer membranes of top-views and side-views of the cells, illustrating further how the Waffle Method may simplify the study of outer membrane-bound proteins.

## Discussion

Cryo-FIB/SEM-ET has shown enormous promise in studying biological systems in three-dimensions at the highest resolution available[2–5]. For instance, the ability to study drug delivery processes at the molecular level in native cells has the potential to change drug development, and the ability to examine disease phenotypes in patient tissue at the individual protein level may alter personalized medicine. Recent developments in sub-tomogram processing[20–25] are enabling molecular analysis of cellular specimens. Several bottlenecks in the cryo-FIB/SEM sample preparation portion of the conventional cryo-FIB/SEM workflow remain before these visions may potentially become routine and mainstream, such as specimen vitrification issues,

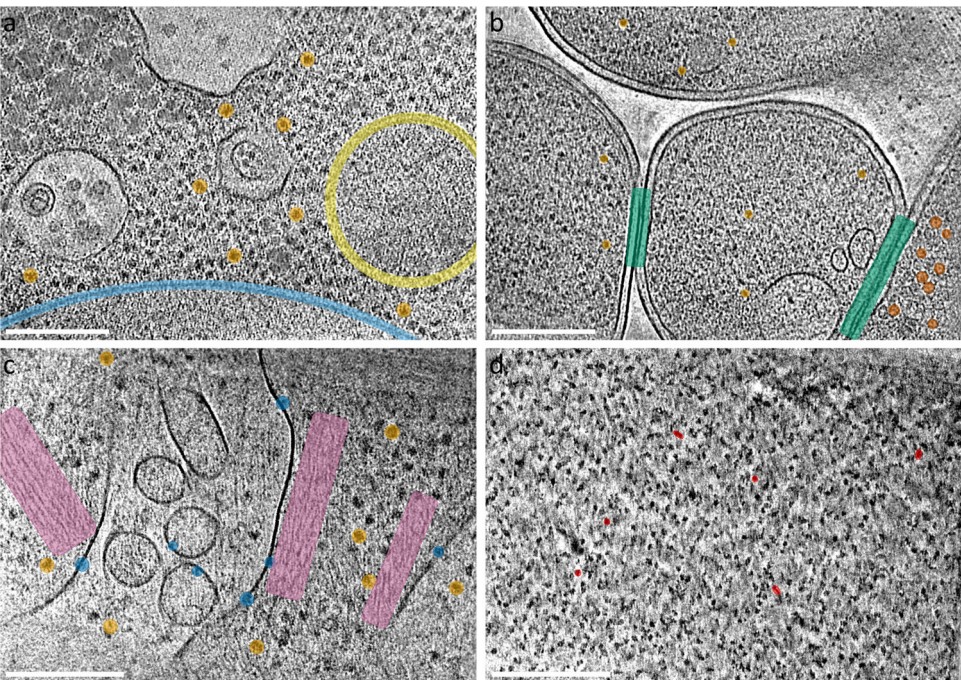

**Fig. 4 Additional examples using the Waffle Method.** Tomogram slice-through of: **a** of yeast *S. cerevisiae* with several ribosomes (orange), potential mitochondria (yellow), and the nuclear envelope (light blue) highlighted; **b** *E. coli* BL21 (DE3) + *Leviviridae* PP7-PP7 virus capsid proteins with several virus capsids (red-orange), ribosomes (orange), and neighboring cell-cell membranes (green) highlighted; **c** HEK 293 S GnTI- with several transmembrane proteins (blue), ribosomes (orange), and actin networks (purple) highlighted; and **d** porcine thyroglobulin (ThG) single particle waffle lamellae with several orientational views highlighted (red). Scale bars are 200 nm. Each slice shown is about 8 Å thick. n > 25 **a**, 35 **b**, 5 **c**, and 10 **d** independent tomograms.

milling specimens with preferred orientation, and low-throughput due to specimen size and/or concentration.

To address these issues, we present the Waffle Method for cryo-FIB/SEM specimen preparation. We illustrated the advantages of the Waffle Method by applying it to microsporidian spores to obtain missing orientations of the spores and the polar tube inside them, and to increase concentration and throughput. We illustrated the broadness of the Waffle Method by applying it to three additional specimens—yeast *S. cerevisiae*, *E. coli*, and HEK 293 S cells—and to different cryo-FIB/SEM hardware. Other directionally-oriented organelles inside the cells may also be studied now with the Waffle Method. Additionally, the Waffle Method may present opportunities for cell-cell interaction studies, as suggested in Fig. 4b. In addition, several colleagues in the field have confirmed to us that Waffle Method is compatible with other hardware, including the Leica ICE HPF and the Zeiss Crossbeam cryo-FIB/SEM.

Waffle Method success required several key components and one key development. Three key components that resolve the critical issue of waffle lamella curtaining are (1) polishing planchette hats which substantially smooths the initial waffle surface, creating sufficiently smooth waffle tops (Supplementary Fig. 1a–e), (2) depositing a sufficient amount of platinum GIS prior to FIB-milling to protect the lamella during milling (Supplementary Fig. 5), and (3) consistently milling lamella with the FIB beam initially intersecting the platinum GIS coat. Other key components for the Waffle Method workflow include: (4) the addition of ~25 nm carbon coat on the EM grid prior to specimen application and freezing, which strengthens the substrate to help reduce broken squares, (5) using 1-hexadecene to allow the planchette hats and optional spacer to be more easily separated after high-pressure freezing (Supplementary Fig. 1f), (6) milling lamellae in parallel to reduce lamellae contamination. Lastly, (7) the development of notch milling was key for substantially increasing the likelihood of lamellae surviving preparation and grid transfers.

Notch milling in particular proved to be a critical part of Waffle Method development. Initially, gap milling similar to Wolff et al.[26] was independently developed and tested prior to any publications (Supplementary Fig. 7), including milling gaps directly into the sides of lamellae, but this did not appreciably increase the survivability of waffle lamellae. Due to the substantial inflexibility of the frozen waffle slab on the grid, any directional or angular forces on the grid are primarily transferred to the lamellae, which destroy them. Subsequently, a new stress-relief milling design, called notch milling, was developed that substantially increases the resiliency of lamellae without causing an increase in lamellae drift when collecting cryo-ET tilt-series. Notches are milled part-way through coarse lamellae milling where one side of the future lamella is completely milled through using the pattern shown in Fig. 2. The notch milling pattern provides freedom of movement for the lamellae to absorb physical and thermal directional and angular forces on the grid in any direction (Supplementary Fig. 8 and Supplementary Movie 9). We hypothesize that notch milling provides support when the grid is not experiencing impulses by providing places for the lamellae to rest at one or more points along the length of the notch. We have observed several instances where the notch tab appears to be safely in contact with the apposed notch (e.g. Supplementary Fig. 9). In our experience, this support is sufficient for lamellae survival during grid transfer and lamellae stability during cryo-ET tilt-series collection.

Currently, we routinely freeze and mill waffle grids using the Aquilos 2 and AutoTEM or the Helios with SerialFIB[27] with high success rates. Within 1 h, we can reliably freeze 4 waffle grids, each of which have multiple squares suitable for milling. Then, during an ~18 h cryo-FIB/SEM session (~6 h of trench milling and waffle lamellae site preparation, followed by ~12 h of automated overnight milling), we can mill about 7 waffle lamellae of size 12 × 12 μm, each 200–250 nm thick. On the second day, we can typically collect dozens of cryo-ET tilt-series from all lamellae. Nearly 100% of waffle lamellae survive both lamellae milling and grid transfers. We find, however, that lamellae occasionally show devitrified areas. Sometimes on a single grid there will be properly vitrified lamellae and lamellae with some devitrified areas. Vitrification solutions are currently being explored.

For cells the size of microsporidia or *E. coli* cells, we estimate that the Waffle Method produces on the order of 100 random cross-sections of cells per lamellae (between 10–20 μm wide), while conventional cryo-FIB/SEM produces a handful with less orientational variation. For a typical day of milling, ~7 such lamellae may be produced by the Waffle Method with semi-automation while ~20 may be produced by conventional cryo-FIB/SEM with full automation. We estimate that the Waffle Method improves throughput by an order of magnitude over conventional cryo-FIB/SEM under these conditions.

Currently, we find that routinely milling lamellae in a semi-automated fashion using AutoTEM or SerialFIB at a size of 12 × 12 μm is optimal for positioning lamellae windows during milling, targeting cryo-ET locations, and for routinely high lamellae survivability. We are also testing milling much larger lamellae, such as the ~70 × 25 μm lamella in Supplementary Fig. 6, to further increase throughput. While this lamella may be too thick for high-resolution analysis, it may still be useful for medium-resolution analysis. We find that as lamella size increases, survivability decreases. We are currently exploring reliable solutions for making such very wide lamellae uniformly ~200 nm thick while still surviving transfer to the cryo-TEM.

There are several optional and alternative steps in the Waffle Method:

- If there are vitrification issues due to the planchette hats not being completely filled with sample prior to high-pressure freezing, then it may be useful to apply 2-methylpentane to the nearly-assembled waffle after adding the sample. After high-pressure freezing, the 2-methylpentane may be sublimed away, leaving a properly vitrified sample as described in Harapin et al.[18].
- If the sample is too viscous, then instead of pipetting the sample onto the grid, the sample may be smeared onto the grid like peanut butter.
- Instead of sputtering carbon onto the EM grid to rigidify it, gold may be sputtered instead.
- Grid spacer(s) may be used, as depicted in Fig. 1 and Supplementary Movie 1, to position the waffle closer to the center of the grid or to attempt to thicken the waffle. Positioning the waffle towards the center of the grid may increase the flexibility of the grid + waffle, which may in turn increase the survivability rate of waffled lamellae. Thickening the waffle may be useful for particularly large specimens, however caution should be taken not to make the waffle thicker than about 50 μm, which is the typical depth of focus for cryo-FIB-milling during fine milling[9]. The ~50 μm thickness limit may be circumvented by first preparing a thicker waffle, trench milling wide trenches at a moderate angle, then milling the bottom off at shallower angles until the remaining slab is less than 50 μm thick, and then finishing milling. This strategy will only leave the top ~50 μm of the slab available for milling, which may limit the tilt angle range during cryo-ET collection if the trenches are not large enough.
- Planchette hats of different sizes (ie. indentation depths) may be used to optimize waffle thickness.

- It may be useful to add a grid to the top of the sample (ie. between the sample and the top planchette in Fig. 1f) during waffle assembly to try to force it to be flat on top.
- The waffle may be assembled outside of the HPF tip. We had the best success preparing it inside the HPF tip as shown in Supplementary Fig. 2 and Supplementary Movie 2.
- Platinum GIS may be applied to the side of freshly-cut trenches to allow for milling lamellae with the leading edge on the side of the trench rather than the top.
- An arrowhead notch (Supplementary Fig. 10) may be used instead of the notch proposed in this manuscript. We have not tested if there is a difference in lamellae survivability depending on which notch type is used.
- A cryo-FLM may be used to optimize localization of objects of interest at any stage during specimen preparation on the grid.
- Cryo-ASV or an in-chamber cryo-FLM may be used concurrently with FIB-milling to help localize objects of interest in the z-direction and to gain sample context.
- A magnifying glass and/or a camera may be used when positioning the grid into the cryo-FIB/SEM holder and the cryo-EM holder for cryo-ET collection to orient the squares to be parallel or perpendicular to the tilt range of the beam as necessary.
- We have explored using 50 mesh and 100 mesh grids in order to increase the size of the lamellae. However, in our experience the grid and lamellae become less stable due to the larger squares.

There are several applications of the Waffle Method that are currently being explored. Figure 5 depicts three potential applications of the Waffle Method. (1) Long complexes like filaments and microtubules may be studied from all directions (Fig. 5a), as opposed to in conventional single particle preparation where axial views do not exist. (2) Single particle proteins may be studied in bulk solution after waffle milling, which will circumvent any issues caused by the air-water interface or substrate-protein interface in conventional single particle preparation, including denaturation, aggregation, preferred orientation, no particles in holes, etc[14–16]. (Fig. 5b). (3) Long, skinny cells, such as the *Spirochaeta plicatilis* depicted in Fig. 5c, may be studied in all orientations. Each of these potential applications will inherently result in partially milled objects that will need to be removed or avoided during processing.

A preliminary experiment suggests that applying the Waffle Method to single particle specimens is possible. Figure 4d and Supplementary Movie 10 shows a tomogram of porcine thyroglobulin (ThG) as a single particle sample from a waffle lamella. The ThG sample, prepared at 100 mg/mL, shows randomly oriented ThG particles that are mono-dispersed in 3D. Based on the bulk sample concentration and the number of observed particles in the tomograms, we estimate that the density of particles in bulk and in the lamellae are roughly the same, suggesting that the Waffle Method and recent cryo-ET processing algorithms[20–22,25] may solve all conventional single particle cryo-EM air-water interface issues. However, further studies in this direction across several specimens are required before definitive conclusions may be made, and the requirement of much higher concentration compared to conventional single particle cryo-EM should be noted.

There are several other issues that the Waffle Method may alleviate. (1) The Waffle Method may be used to study cells that do not behave on conventionally-prepared EM grids due to

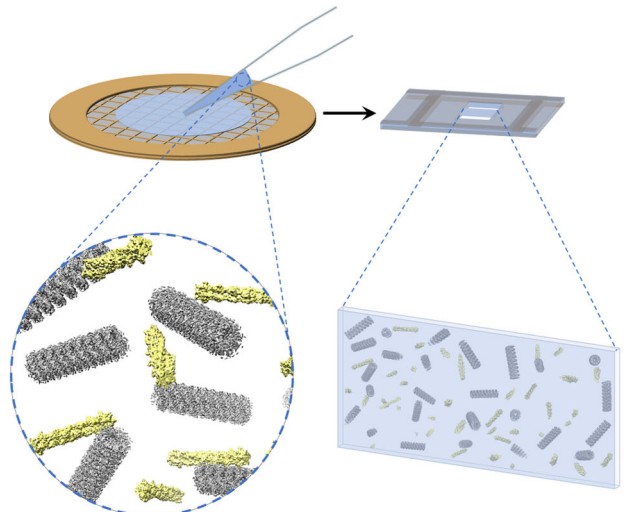

## a) Thin, long complexes

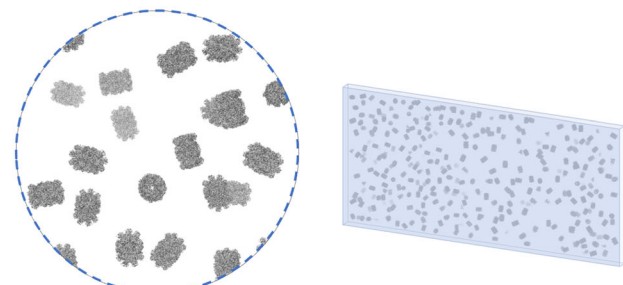

## b) Single particles

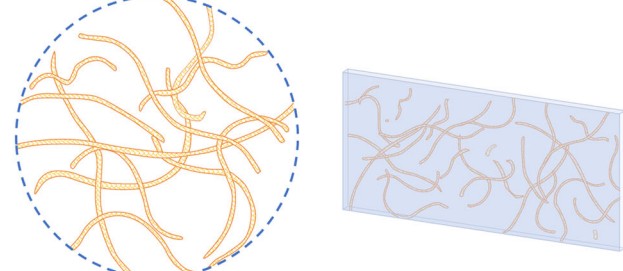

## c) Long cells

**Bulk sample**      **Waffled lamella**

**Fig. 5 Additional potential issues that the Waffle Method may rectify. a** Thin, long complexes, such as actin and microtubules depicted here, will exist in all orientations in waffled lamellae. **b** Single particle protein samples (depicted: internal T20S proteasome EM density map) that exhibit inhibitive issues (e.g. denaturation, aggregation, preferred orientation) when prepared conventionally in a thin film may be prepared without those issues using the Waffle Method. **c** Long cells, such as the *Spirochaeta plicatilis* depicted here (0.2–0.75 μm in diameter by 5–250 μm in length), may be imaged in all orientations with the Waffle Method, which may also allow for the cell membrane to be studied more thoroughly. UCSF Chimera[41] was used for molecule depiction in (a) and (b).[1].

suboptimal grid affinity, suboptimal concentration in grid squares, wicking/hydration issues, etc. Suboptimal grid affinity may occur because some cells prefer to adhere to grid bars, for example. Suboptimal concentration in grid squares may be defined as sample concentration differences between the time before applying the sample to the grid and after freezing. Wicking/hydration issues on grids include cells that are so dry that the biological significance is impacted and grids that are so wet that the cells are difficult to locate after freezing or that are not properly vitrified. (2) Due to the random location and orientation of cells in waffles lamellae, there will be cross-sections of outer cell membranes in waffles (Supplementary Movie 8), which are difficult or impossible to obtain with conventional cryo-FIB/SEM. (3) Tissues and organisms may be studied using the Waffle Method instead of performing cryo-FIB/SEM liftout.

Compared to cryo-liftout, the Waffle Method may have several benefits. (1) Less specialized hardware is required; liftout requires an in-chamber tool, separate grid holder, and different grids. (2) Less operator specialization is required. (3) Higher-throughput may be achieved; more lamellae can be made per grid. (4) Finally, the Waffle Method may be more automatable due to all cryo-FIB/SEM operations being performed on one grid.

Here, we presented three waffled specimens that took advantage of the Aquilos 2 AutoTEM lamellae milling automation software. Specifically, the steps that we automated were coarse-to-fine milling and polishing (Supplementary Movie 4). Currently, waffle lamellae locations must first be manually trench milled and manually notch milled before automated lamellae coarse-to-fine milling can be performed in AutoTEM or SerialFIB due to the current limitations of the softwares. We anticipate that all waffle milling steps will be automated in the future.

The Waffle Method may not provide all of the described benefits to adherent cells that stop replicating after confluence and do not form multiple layers. However, these types of samples may benefit from proper vitrification due to high-pressure freezing and from the large lamellae that waffle milling produces (conventional cryo-FIB/SEM cannot provide as large lamellae, even for clumped cells, because the top-surface of the cells is not topologically uniform). However, the lamellae will be mostly empty except for the bottom of the lamellae that will contain slices of cells as desired. The development of grids with shorter grid bars may be used to optimize waffle thickness. As a result, throughput may be slightly increased compared to conventional cryo-FIB/SEM.

We anticipate that the Waffle Method will broaden the specimens amenable to cryo-FIB/SEM while providing good vitrification for specimen of all sizes solving preferred orientation issues, increasing throughput during both cryo-FIB/SEM and cryo-ET, allowing small and skinny cells to be studied, and allowing cells that behave poorly on grids to be studied. We also envision that the Waffle Method may be used as a platform for further method development in cryo-FIB/SEM and cryo-EM. The Waffle Method may also become preferred over cryo-liftout, particularly for tissue specimens, due to the relatively reduced hardware and expertise requirements. We anticipate that the Waffle Method will be fully automated to increase throughput and reduce operator time.

## Methods

**General**. Several variations of the Waffle Method for the different specimens and hardware in this manuscript are described here. We note that implementation of the Waffle Method with new specimens and hardware may require a new combination of these methods.

**Microsporidian spore sample preparation**. The *A. algerae* microsporidian spores shown in Fig. 3b were propagated in *Helicoverpa zea* larvae and purified using a continuous Ludox gradient, as previously described[28]. *E. hellem* microsporidian

spores (ATCC 50504) were used in Fig. 3a,c–g. The spores were propagated in Vero cells (ATCC CCL-81). Vero cells were maintained in Eagle's Minimum Essential Medium (EMEM) (ATCC 30–2003) with 10% heat-inactivated fetal bovine serum (FBS) at 37 °C with 5% $CO_2$. At ~90% confluence, the media was replaced with EMEM supplemented with 3% FBS and the parasites were added into a 25 cm² tissue culture flask. Medium was changed every 2 days. After 14 days post-infection, the cells were detached from the flask using a cell scraper and centrifuged at $1300 \times g$ for 10 min at room temperature. Cell pellets were resuspended with 5 mL of distilled water and mechanically disrupted using a G27 needle. To purify microsporidian spores, 5 mL of a 100% Percoll was added and vortexed prior to centrifugation at $1800 \times g$ for 30 min at 25 °C. The purified spore pellets were washed 3 times with 1X PBS and stored at 4 °C until use. Spore concentration ($1.54 \times 10^8$ spores/mL) was measured using a hemocytometer. Prior to waffle sample application, the spores were centrifuged, and resuspended in a small amount of buffer, resulting in a highly concentrated sample for application to the waffle grid assembly.

**Microsporidian spore waffle grid preparation**. The general, overall Waffle Method workflow is described in the Results section. Here and in the other waffle preparation sections for the other specimen, we describe the specific equipment and workflows used during waffle milling. This section on microsporidian spore waffle grid preparation covers the period of Kotaro's Waffle Method development, and is accurate to the best of our knowledge. Waffle Method development resulted in the following protocol.

We use Quantifoil carbon 200 mesh EM grids (Quantifoil, Jena, Germany). About 25 nm of carbon is sputtered onto the non-grid bar side of the grid using a Leica EM ACE600 High Vacuum Sputter Coater (Leica Biosystems, Germany). Solid brass planchette hats (3.3 mm diameter) are polished for several minutes using POL Metallpflege metal-polish and a Kimwipe. Excess is wiped off using a Kimwipe. The planchette hats and grid spacer are coated with 1-hexadecene. The grid is plasma cleaned using a Gatan Solarus (Gatan Inc, Pleasanton, CA) with the following recipe: 80% $O_2$ gas and 20% $H_2$ gas for 30 s.

The HPF tip is cleaned with ethanol prior to assembling the bottom of the waffle in the HPF tip as shown in Supplementary Movie 2. Several microliters of sample (typically 3–6 μL) are applied to the bottom waffle assembly on the grid bar side such that no air pockets are present, and optionally 2-methylpentane is applied to the top of the sample just prior to placing the top planchette hat on and closing the HPF tip. The assembled waffle grid in the HPF tip is quickly transferred to a Wohlwend HPF Compact 01 high-pressure freezer where it is high-pressure frozen. The HPF tip is then disassembled to release the waffled grid inside of the planchette hats. The planchette hats are disassembled using a combination of tweezers, flat screwdrivers, and razor blades to reveal the waffled grid. The waffle grid is then carefully clipped.

The grid is placed in a flat or tilted cryo-FIB/SEM holder and inserted into the Quorum PP3000T prep chamber (Quorum Technologies, Great Britain) attached to the FEI Helios NanoLab 650 cryo-FIB/SEM (FEI, Hillsboro, OR) at NYSBC. Conductive platinum is sputtered onto the waffled grid inside the prep chamber for 60 s before placing the shuttle onto the cold stage in the main chamber. Platinum GIS is deposited onto the waffle for 4–7 s at 35 °C at 7 mm from working distance. Grid bar lines are drawn into the waffle to help locate areas of interest and orient the cryo-FIB/SEM operator. Two trenches tens of microns in dimensions and about 30 μm apart for each area of interest are milled at between 45° and 90° from the grid plane with a milling current between 9.3 nA and 0.79 nA. If trench milling is performed far from 90°, the tilt angle is decreased several times by 5° while milling the bottom area under the slab to ensure that no material remains. The grid is placed into a tilted holder, if not already in one, and placed back into the cryo-FIB/SEM chamber. Eucentric height is obtained for each trenched slab location and positions are saved at shallow angles below 20° from the grid plane. All subsequent milling is performed in parallel* across all lamellae and with the FIB beam first intersecting the platinum GIS layer** to minimize curtaining. The slabs are coarse milled with a milling current between 2.5 nA and 0.43 nA until the slab is about 3 μm thick. A notch is milled into one side of the slab as described in the main text and as shown in Fig. 2. Current is reduced to 0.23 nA and the slab is milled down to 1.5 μm thick with a tab left on the lamella beside the notch while tilting the stage ±1° and while milling more on the carbon side than the sample side. Current is reduced to 80 pA (using the cleaning cross section (CCS) pattern type) and the lamella is milled down to 0.5 μm thick while tilting the stage ±0.5°. With current still at 80 pA CCS and without tilting, the lamella is milled down to the desired final thickness (usually 200 nm or less) before polishing. The grid is removed from the prep chamber and conductive platinum is sputtered onto the grid and lamellae for several tens of seconds.

*Milling in parallel means to first magnify to the location of slab 1 for trench milling, slab 2 for trench milling, slab 3 …, then slab 1 for coarse milling, slab 2 for coarse milling, slab 3 …, then lamella 1 for fine milling, lamella 2 for fine milling, lamella 3 …, and finally lamella 1 for polishing, lamella 2 for polishing, lamella 3 …

**After trench milling, two surfaces are exposed for coarse and fine milling: (1) The frozen sample and (2) the platinum above the sample. All coarse and fine milling should first hit (2) the platinum above the sample. This will minimize curtaining.

**Microsporidian spore Cryo-ET collection**. Tilt-series were collected with Leginon[29] using a TFS Titan Krios (Thermo Fisher Scientific) using counting mode on a Gatan K2 BioQuantum with the energy filter slit width set to 20 eV (NYSBC Krios #3). Tilt-series were collected with a nominal defocus of −6 μm, pixelsize of 3.298 Å, 1.13–1.74 e-/Å² dose per tilt image where dose was increased with the cosine of the tilt angle resulting in 15–23 frames per tilt image and 68 e-/Å² total dose per tilt-series. Collection was performed bi-directionally from [0:50]° to [0:−50]° with 2° tilt increments. A total of 27 tilt-series were acquired.

**Microsporidian spore Cryo-ET processing**. Tilt images were frame aligned with MotionCor2[30] without patches or dose weighting. Frame aligned tilt images were used for fiducial-less tilt-series alignment in Appion-Protomo[31–33]. Only tilt images from [−50:20]° were used for the tomogram in Fig. 3e due to excessive stage drift. All tilt images were used for the right tomogram in Fig. 3f. Tilt-series were dose weighted in Appion-Protomo using equation 3 in Grant & Grigorieff[34] prior to reconstruction with Tomo3D[35,36] SIRT, then denoised using the Topaz-Denoise[37] pre-trained model. Tomograms were visualized with IMOD[38].

**Microsporidian conventional cryo-FIB/SEM grid preparation**. In total 3 μL of *Anncaliia algerae* spores (6.8 × 10⁷ spores/mL) was applied to a Quantifoil holey carbon grid on copper support (2/2, 400 mesh) glow discharged for 1 min, and back-blotted for 15 s. Then, the sample was frozen in liquid ethane using the Leica EM GP plunge freezer. The grid was screened on an FEI Talos Arctica cryo-TEM at the cryo-EM Shared Resource at NYU School of Medicine. The FEI Helios NanoLab 650 at NYSBC was used to perform conventional cryo-FIB/SEM. Briefly, the sample was platinum coated in the Quorum stage, a couple microns of platinum GIS was applied in the Helios main chamber, and spores were individually milled at shallow angles (<15°).

**Yeast cells sample preparation**. The yeast strain *Saccharomyces cerevisiae* YOL058W was grown on solid YPD media at 30 °C. Cells were harvested from the plate and centrifuged in liquid YPD media for 4 min at ~4,200 × g at 4 °C. The cell pellet was resuspended just prior to grid preparation.

**Yeast cells waffle grid preparation**. The waffle grid was prepared in the same manner as the 'Microsporidian spore waffle grid preparation' section, except for the following differences. The flat side of the planchette hats (3.3 mm diameter) were sanded down using 1200, 7000, then 15,000 grit sandpaper until smooth. POL Metallpflege metal-polish was applied to the flat side of the hats and polished on filter paper, then any excess polish was wiped off using a Kimwipe. Immediately before assembling, both planchette hats were rinsed with 1-hexadecene. The waffle grid was assembled in the HPF tip as follows: planchette hat (flat side up), glow-discharged Quantifoil 2/2 200 mesh Cu grid with 25 nm extra carbon on the film layer (copper grid bar side up), 5 μL of resuspended cell pellet, planchette hat (flat side down). Excess sample was wiped off with Kimwipes. This assembly was then quickly inserted into a Wohlwend HPF Compact 01 and high-pressure frozen. The waffle was then disassembled under liquid nitrogen and clipped into an autogrid.

**Yeast cells waffle milling**. The waffle grid was placed into a 40° pre-tilted cryo-FIB/SEM grid holder (Leica Microsystems) and inserted into the FEI Helios NanoLab 650 FIB/SEM (Thermo Fisher Scientific) with Leica EM VCT500 cryo-stage at NYSBC. A grid overview image was taken and centrally-located squares with an even surface were selected. The grid was then coated with platinum GIS for 1 min. The grid was then rotated 180° (175° to −5°) and tilted to 60° to mill at a 48° angle relative to the ion beam. Two rectangle trenches were milled at 9.3 nA with a spacing of 15 μm. The grid was then rotated back to the original position and tilted to a milling angle of 5°. Any visible bulk material was removed at 2.5 nA prior to thinning. Each lamella was milled manually following the parameters in Table 1. The first step of the relief notch was milled at 0.23 nA when the lamella was 2 μm in thickness and the final step at 1 μm. Another clean-up of excess material was performed at 2.5 nA at 25° stage tilt prior to the final polishing steps.

**Yeast cells cryo-ET collection**. Tilt-series were collected with Leginon[29] using a TFS Titan Krios (Thermo Fisher Scientific) with a Cs corrector and using counting mode on a Gatan K3 BioQuantum with the energy filter slit width set to 20 eV (NYSBC Krios #2). Tilt-series were collected with a nominal defocus of −4 μm, pixelsize of 2.077 Å, 4.15–6.23 e-/Å² dose per tilt image where the dose was increased with the cosine of the tilt angle resulting in 12–18 frames per tilt image and 180 e-/Å² total dose per tilt-series. The collection was performed bi-directionally from [0:48.6]° then [0:−48.6]° with 2.7° tilt increments. A total of 30 tilt-series were acquired.

**Yeast cells cryo-ET processing**. Tilt images were frame aligned with MotionCor2[30] without patches or dose weighting. Frame aligned tilt images were used for fiducial-less tilt-series alignment in Appion-Protomo[31–33]. Tilt-series were dose weighted in Appion-Protomo using equation 3 in Grant & Grigorieff[34] prior to reconstruction with Tomo3D[35,36] SIRT, then denoised using the Topaz-Denoise[37] pre-trained model. Tomograms were visualized with IMOD[38].

**E. coli sample preparation**. *E. coli* BL21 (DE3) cells were harvested by centrifugation at 6000 × g 4 h post-induction of *Leviviridae* PP7-PP7 protein production as previously described[39]. The resulting cell pellet was resuspended in equal volume of phosphate-buffered saline buffer (PBS) and kept at 4 °C for ~48 h during shipment from Georgia Tech to NYSBC.

**E. coli waffle grid preparation**. The waffle grid was prepared in the same manner as the 'Microsporidian spore waffle grid preparation' section, except for the following differences. The flat side of the planchette hats (3.3 mm diameter) were polished using following workflow: (1) Rub with coarse sandpaper (grit 1200) to remove machining marks from the hats, (2) Rub with fine sand paper (grit 15,000) (Supplementary Fig. 1a–e), and (3) Final polish for 10 s with POL Metallpflege metal-polish and a Kimwipe. Immediately before assembling, both planchette hats were rinsed with 1-hexadecene. The waffle grid was assembled in the HPF tip as the following: planchette hat (flat side up), prepared Quantifoil 2/2 200 mesh Cu grid with 25 nm extra carbon on the film layer (copper grid bar side up), 5 μL of induced *E. coli* BL21 (DE3) sample, planchette hat (flat side down). Any excess sample that spilled out of the assembly was wiped off with a Kimwipe. This assembly was then quickly inserted into a Wohlwend HPF Compact 01 and high-pressure frozen. The waffle was then disassembled under liquid nitrogen and clipped into a notched autogrid.

**E. coli waffle milling**. The waffle grid was placed into a 45° pre-tilted cryo-FIB/SEM shuttle and inserted into the Aquilos 2 cryo-FIB/SEM (Thermo Fisher Scientific) at NYSBC. A mosaic of the grid was taken in TFS MAPS 3.16, and squares with an even surface were visually identified. A conductive platinum layer was then sputtered on the grid for 15 s at 30 mA and 10 Pa. The grid was then put in deposition position and coated with platinum GIS for 2 min. The grid was then rotated 180° (−71.9° to 108.1°) and tilted to 7° to make the grid surface orthogonal to the ion beam. Two trenches were milled at 15 nA with a spacing of 25 μm. The top trench (closer to the grid notch) was 22 μm wide and 37 μm long, while the bottom trench (further from the grid notch) was 20 μm wide and 17 μm long. The grid was then rotated back to the original position and tilted to a milling angle of 26°. The trench closest to the notch was then milled again at this angle to clean up bulk material below the lamellae.

Using AutoTEM Cryo 2.2, the preparation step, which includes eucentric tilting, setting the milling angle, collecting a reference image, and lamella placement, is done for each lamella site. Another cleanup of the trench closer to the grid notch, at 3 nA, and a relief notch, at 0.1 nA and about 200 nm wide, is milled at an angle of 20°. Each lamella placement is then repositioned about 1 μm from the relief notch. Each lamella is then milled and thinned stepwise using the parameters in Table 2.

**E. coli cryo-ET collection**. Tilt-series were collected with Leginon[29] using a TFS Titan Krios (Thermo Fisher Scientific) with a Cs corrector and using counting mode on a Gatan K3 BioQuantum with the energy filter slit width set to 20 eV (NYSBC Krios #2). Tilt-series were collected with a nominal defocus of −4 to −8 μm, pixelsize of 2.077 Å, 4.87–9.39 e-/Å² dose per tilt image where dose was increased with the cosine of the tilt angle resulting in 14–27 frames per tilt image and 234 e-/Å² total dose per tilt-series. Collection was performed bi-directionally from [0:48]° then [0:−58]° with 2.8° tilt increments. A total of 41 tilt-series were acquired.

### Table 1 Manual FIB-milling and thinning parameters on the Helios cryo-FIB/SEM.

| | Lamella thickness (μm) | Milling angle | Milling Current (nA) | Pattern type |
|---|---|---|---|---|
| Rough Milling | 3 | 17° | 0.79 | Rectangle |
| Medium Milling | 2 | 17° | 0.43 | Rectangle |
| Fine Milling | 1.5 | 17° | 0.23 | Rectangle |
| Over/Under-tilt | 1 | 16.5°, 17.5° | 0.23 | CCS |
| Polishing 1 | 0.30 | 17° | 0.080 | CCS |
| Polishing 2 | 0.20 | 17° | 0.024 | CCS |

**Table 2 AutoTEM Cryo 2.2 FIB-milling and thinning parameters on the Aquilos 2.**

|  | Lamella thickness (defined as "Pattern Offset") (µm) | Front Width Overlap (µm) | Rear Width Overlap (µm) | Milling Current (nA) | Pattern Type | DCM Rescan Interval (s) |
|---|---|---|---|---|---|---|
| Rough Milling | 1.0 | 1.5 | 1.0 | 1.0 | Rectangle | 120 |
| Medium Milling | 0.8 | 0.65 | 0.5 | 0.5 | CCS | 90 |
| Fine Milling | 0.6 | 0.35 | 0.1 | 0.3 | CCS | 60 |
| Finer Milling | 0.4 | 0.05 | 0.05 | 0.3 | CCS | 30 |
| Polishing 1 | 0.15 | N/A | N/A | 0.03 | CCS | 30 |
| Polishing 2 | 0 | N/A | N/A | 0.01 | CCS | 10 |

Milling and thinning were done at 20°, and a depth correction of 100% and 0° overtilt at 30 kV while polishing steps were set to 160%.

**E. coli cryo-ET processing**. Tilt images were frame aligned with MotionCor2[30] without patches or dose weighting. Frame aligned tilt images were used for fiducial-less tilt-series alignment in Appion-Protomo[31–33]. Tilt-series were dose weighted in Appion-Protomo using equation 3 in Grant & Grigorieff[34] prior to reconstruction with Tomo3D[35,36] SIRT, then denoised using the Topaz-Denoise[37] pre-trained model. Tomograms were visualized with IMOD[38].

**HEK 293 S cells sample preparation**. Suspension HEK 293 S GnTI⁻ cells (ATCC) were cultured in the Freestyle 293 expression medium (GIBCO) at 37 °C and 5% $CO_2$. A 10 mL aliquot was centrifuged for 5 min at 500 × g at room temperature, then the resulting cell pellet was resuspended to a final concentration of 3.5e8 in culturing media.

**HEK 293 S cells waffle grid preparation**. The waffle grid was prepared in the same manner as the 'Microsporidian spore waffle grid preparation' section, except for the following differences. The flat side of the planchette hats (3.3 mm diameter) were sanded down using 1200, 7000, then 15,000 grit sandpaper until smooth. POL Metallpflege metal-polish was applied to the flat side of the hats and polished on filter paper, then any excess polish was wiped off using a Kimwipe. Immediately before assembling, both planchette hats were rinsed with 1-hexadecene. The waffle grid was assembled in the HPF tip as follows: planchette hat (flat side up), glow-discharged gold Quantifoil grid with 20 nm of extra carbon on the film layer (copper or gold grid bar side up), 4 µL of resuspended cell pellet, planchette hat (flat side down). Excess sample was wiped off with blotting paper #4. This assembly was then quickly inserted into a Wohlwend HPF Compact 01 and high-pressure frozen. The waffle was then disassembled under liquid nitrogen and clipped into a notched autogrid.

**HEK 293 S cells waffle milling**. The waffle grid was placed into a 45° pre-tilted cryo-FIB/SEM shuttle and inserted into the Aquilos 2 cryo-FIB/SEM (Thermo Fisher Scientific) at NYSBC. A grid overview was taken in MAPS 3.16, and squares subjectable for milling were targeted as Lamella sites. A conductive platinum layer was then sputtered on the grid for 15 s at 30 mA and 10 Pa. The grid was then put in deposition position and coated with platinum GIS for 2 min. The grid was then rotated 180° (actual angles −71.9° to 108.1°), tilted to 7° to make the grid surface orthogonal to the ion beam. Two trenches were milled at 15 nA with a spacing of 25–30 µm. The top trench (closer to the grid notch) was 22 µm wide and 32–37 µm long, while the bottom trench (further from the grid notch) was 20 µm wide and 17 µm long. The grid was then rotated back to the original position. Using AutoTEM Cryo 2.2, the "Preparation" step is done in guided mode for each lamella site, which includes eucentric tilting, setting the milling angle (was set to 20°), collecting a reference image, and an initial lamella placement. Then, bulk material below the lamellae were cleaned up at 7 nA (at maximum stage tilt angle) and 3 nA (at milling angle) until there was no visible leftover material left. After the cleanup, a notch about 200 nm wide was added at the milling angle at 0.3 nA. The total length of the notch was about 5 um. Ultimately, the image acquisition and lamella placement steps in "Preparation" were repeated for each site while repositioning the lamella about 1 µm away from the milled notch. Each lamella is then milled and thinned with AutoTEM in stepwise mode using the parameters from Table 2.

**HEK 293 S cells cryo-ET collection**. Tilt-series were collected with Leginon[29] using a TFS Titan Krios (Thermo Fisher Scientific) with a Cs corrector and using counting mode on a Gatan K3 BioQuantum with the energy filter slit width set to 20 eV (NYSBC Krios #2). Tilt-series were collected with a nominal defocus of −4 to −5 µm, pixelsize of 1.632 Å, 4.12–5.83 e-/Å² dose per tilt image where dose was increased with the cosine of the tilt angle resulting in 12–17 frames per tilt image and 180 e-/Å² total dose per tilt-series. Collection was performed bi-directionally from [0:59]° then [0:−39]° with 2.8° tilt increments. A total of 6 tilt-series were acquired.

**HEK 293 S cells cryo-ET processing**. Tilt images were frame aligned with MotionCor2[30] without patches or dose weighting. Frame aligned tilt images were

used for fiducial-less tilt-series alignment in Appion-Protomo[31–33]. Tilt-series were dose weighted in Appion-Protomo using equation 3 in Grant & Grigorieff[34] prior to reconstruction with Tomo3D[35,36] SIRT, then denoised using the Topaz-Denoise[37] pre-trained model. Tomograms were visualized with IMOD[38].

**ThG sample preparation**. Porcine thyroglobulin from porcine thyroid gland (Sigma T1126) was dissolved in 50 mM HEPES pH 7.4 and 100 mM NaCl. The final concentration was about 100 mg/ml.

**ThG waffle grid preparation**. The waffle grid was prepared in the same manner as the 'Microsporidian spore waffle grid preparation' section, except for the following differences. The flat side of the planchette hats (3.3 mm diameter) were polished using following workflow: (1) Rub with coarse sandpaper (grit 1200) to remove machining marks from the hats, (2) Rub with fine sand paper (grit 15,000), and (3) Final polish for 10 s with POL Metallpflege metal-polish and a Kimwipe (Supplementary Fig. 1). Immediately before assembling, both planchette hats were rinsed with 1-hexadecene. The waffle grid was assembled in the HPF tip as the following: planchette hat (flat side up), prepared Quantifoil 2/2 200 mesh Cu grid with 25 nm extra carbon on the film layer (copper grid bar side up), 5 µL of 100 mg/ml porcine Thyroglobulin sample, planchette hat (flat side down). Any excess sample that spilled out of the assembly was wiped off with a Kimwipe. This assembly was then quickly inserted into a Wohlwend HPF Compact 01 and high-pressure frozen. The waffle was then disassembled under liquid nitrogen and clipped into a notched autogrid.

**ThG waffle milling**. The waffle grid was placed into a 45° pre-tilted cryo-FIB/SEM shuttle and inserted into the Aquilos 2 cryo-FIB/SEM (Thermo Fisher Scientific) at NYSBC. A mosaic of the grid was taken in TFS MAPS 3.16, and squares with an even surface were visually identified. A conductive platinum layer was then sputtered on the grid for 15 s at 30 mA and 10 Pa. The grid was then put in deposition position and coated with platinum GIS for 2 min. The grid was then rotated 180° (−71.9° to 108.1°) and tilted to 7° to make the grid surface orthogonal to the ion beam. Two trenches were milled at 15 nA with a spacing of 25 µm. The top trench (closer to the grid notch) was 22 µm wide and 37 µm long, while the bottom trench (further from the grid notch) was 20 µm wide and 17 µm long. The grid was then rotated back to the original position and tilted to a milling angle of 26°. The trench closest to the notch was then milled again at this angle to clean up bulk material below the lamellae.

Using AutoTEM Cryo 2.2, the preparation step, which includes eucentric tilting, setting the milling angle, collecting a reference image, and lamella placement, is done for each lamella site. Bulk material below the lamella was cleaned at 7 nA (at maximum stage tilt angle) then at 3 nA (at a milling angle of 20°). After, a relief notch is added, at 0.3 nA and about 200 nm wide (at a milling angle of 20°). Each lamella placement is then repositioned about 1 µm from the relief notch. Each lamella is then milled and thinned stepwise using the parameters in Table 2.

**ThG cryo-ET collection**. Tilt-series were collected with Leginon[29] using a TFS Titan Krios (Thermo Fisher Scientific) with a Cs corrector and using counting mode on a Gatan K3 BioQuantum with the energy filter slit width set to 20 eV (NYSBC Krios #2). Tilt-series were collected with a nominal defocus of −5 µm, pixelsize of 1.632 Å, 3.34–6.67 e-/Å² dose per tilt image where dose was increased with the cosine of the tilt angle resulting in 9–18 frames per tilt image and 148 e-/Å² total dose per tilt-series. Collection was performed bi-directionally from [0:45]° then [0:−60]° with 3° tilt increments. A total of 15 tilt-series were acquired.

**ThG cryo-ET processing**. Tilt images were frame aligned with MotionCor2[30] without patches or dose weighting. Frame aligned tilt images were used for fiducial-less tilt-series alignment in Appion-Protomo[31–33]. Tilt-series were dose weighted in Appion-Protomo using equation 3 in Grant & Grigorieff[34] prior to reconstruction with Tomo3D[35,36] SIRT, then denoised using the Topaz-Denoise[37] pre-trained model. Tomograms were visualized with IMOD[38].

**Reporting summary**. Further information on research design is available in the Nature Research Reporting Summary linked to this article.

## Data availability

The *E. coli* raw frame data generated in this study have been deposited in the Electron Microscopy Public Image Archive (EMPIAR) database under accession code EMPIAR-10981. One of the tomograms from this full dataset (Fig. 4b) has been deposited in the Electron Microscopy Data Bank (EMDB) database under the accession code EMD-26396.

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

## Acknowledgements

This manuscript is dedicated to Kotaro Kelley and his family to serve as a memory of him. Kotaro conceived of and developed the Waffle Method beginning in 2018, but passed away before this manuscript was written and the method could be reported on in the literature. Kotaro conceived of the Waffle Method as a specific solution to several collaborative projects, which he quickly realized was a general solution to a broad range of issues in the field of cryo-FIB/SEM. Kotaro's work from 2018 to 2019 was published as a preprint[40], followed by our re-discovery of his method. Kotaro spent countless nights and weekends of trial and error developing the Waffle Method. As Kotaro was the only Waffle Method expert during microsporidia waffle preparation, we present here as much knowledge and data for these portions of the manuscript as we have access to and that we understand. This manuscript in part serves as a method to disseminate the knowledge and skills developed by Kotaro before he passed. The Waffle Method examples in Fig. 3 and Supplementary Figs. 4b–f, 5, 7 were all performed by Kotaro. We also report on Kotaro's independent exploration of stress-relief gaps for lamella. We are continuing to develop the Waffle Method and extensions of it as described in the Discussion. Here we report on the Waffle Method in lieu of Kotaro to recognize his work and honor his dedication; this manuscript serves as a time capsule of his work in these areas. We hope Kotaro's knowledge and spirit continue to make positive impacts on people's lives through biomedical research that extend beyond and are derivatives of his research. We thank James J. Becnel and Neil Sanscrainte for the microsporidian *A. algerae* sample and Nicolas Coudray for helping screen the grid at the cryo-EM Shared Resource at NYU School of Medicine. We thank M.G. Finn and Liangjun Zhao from Georgia Tech for providing *E. coli* BL21 cells expressing PP7-PP7 virus-like particles. We thank Sagar Khavnekar at Max Planck Institute of Biochemistry (MPIB) and Dr. Philipp Erdmann at MPIB (now at Human Technopole) for sharing their experiences with implementing the Waffle Method. We thank Viacheslav Serbynovskyi for testing potential alternative Waffle Method freezing techniques. We thank Alexander I. Sobolevsky and Maria V. Yelshanky for providing the HEK 293 S cells. A.J.N. was supported by a grant from the NIH National Institute of General Medical Sciences (NIGMS) (F32GM128303). P.J. was supported by the American Heart Association (19POST34430065). G.B. was supported by Pew Biomedical Scholars (PEW-00033055), Searle Scholars Program (SSP-2018-2737) and the National Institute of Allergy and Infectious Diseases (R01AI147131). Some of this work was performed at the Simons Electron Microscopy Center located at the New York Structural Biology Center, supported by grants from the Simons Foundation (SF349247), NIH NIGMS (GM103310), and NIH (U24GM139171).

## Author contributions

K.K. conceived of this project, developed and implemented the Waffle Method, and prepared and collected the microsporidia waffle cryo-FIB/SEM-ET data. K.K. independently conceived and tested gap milling methods and notch milling. A.J.N. prepared, collected, and analyzed the conventional cryo-FIB/SEM data. P.J. prepared the microsporidian spore samples. K.K. and A.J.N. processed and analyzed the microsporidia waffle data. A.M.R., O.K., D.B., and M.K. prepared, waffled, and collected the yeast, *E. coli*, and HEK 293 S cells, and A.J.N. processed and analyzed the tomograms. C.S.P conceived of cryo-FIB milling bulk single particle samples. K.K., A.M.R., O.K., P.J., D.B., M.K., E.E., G.B., C.S.P., B.C., and A.J.N. designed the experiments. A.J.N. wrote the manuscript, created the figures, and created the movies and animations. A.M.R., O.K., P.J., D.B., M.K., E.E., G.B., C.S.P., B.C., and A.J.N. edited the manuscript.

## Competing interests

The authors declare no competing interests.
