## [Peer Review File · Nature Communications]

Waffle Method: A general and flexible approach for improving throughput in FIB-millingReviewers' Comments:

Reviewer #1:

Remarks to the Author:

The manuscript summarizes a new approach to make a wider range of samples amenable for focused ion beam milling and cryo electron tomography, without the need for extra lift-out hardware. The Waffle method described by the authors has several advantages to other methods, such as: samples of tens of microns in thickness can be vitrified and processed, and there is no issue with preferred orientation of certain samples on the support grid. The authors provide detailed instructions on how to use the method and provide multiple convincing examples. With even more automation coming to the field, this method can soon become even more accessible and streamlined.

The Waffle method described here is very elegant and is of wide interest to the community.

A few minor comments:

- Could the authors please, provide one more figure summarizing all the main steps from having the waffled sample in the FIB-SEM and until the polished lamellae? A perfect figure (in my view) would have a column with schematic drawings (with average dimensions of lamellae/cuts/notches and their geometries), a column with actual example images, and some text next to them to describe each step. This would be a great aid for the future user of the method. Potentially the information from table 1/2 can be added to such a figure.

- Could the authors add typical dimensions of the stress-relief notch that they used to the Supp. Figure 7? The length, height, width, in order for readers to reproduce this closely. Please, also add which currents were used for notch milling to the Table 1.

Reviewer #2:

Remarks to the Author:

This manuscript by Kelley et al. presents a new workflow, named the Waffle method, for cryo-electron tomography (cryo-ET) sample preparation. The Waffle method is based on the use of high-pressure freezing applied to samples attached to, or suspensions deposited on, standard electron microscopy grids. After freezing, these grids, transferred into a cryo-scanning microscope equipped with a focused ion beam (FIB) and thin vitreous lamellae, are prepared under cryogenic conditions. Tomographic data are then recorded from these lamellae using a transmission cryo-electron microscope.

The Waffle method combines important findings that enable researchers to overcome several significant limitations of the existing cryo-lamellae preparation approaches. The use of high-pressure freezing allows the Waffle method to be applied to samples that cannot be reliably vitrified by plunge freezing, which is commonly used to fabricate cryo-lamellae from flat cells growing on, or attached to, EM grids. Thus, the approach proposed by the authors can be applied to a wide range of samples, such as cells, tissues, and embryos. On the other hand, the authors have demonstrated the potential of the method to solve orientation and denaturation problems related to the water-air interface in single-particle reconstruction methods. Importantly, the Waffle method makes it possible to prepare the lamellae directly on the sample grid, which helps to achieve higher throughput and does not require expensive equipment, such as the lift-out method, for samples a few tens of microns thick. Additionally, the authors have proposed an arrow-shaped notch geometry, which is an interesting addition to existing developments for mechanical lamella stabilization.

To summarize, the methodological developments presented in this manuscript are of great importance for the further development of in situ structural analysis approaches and will be enthusiastically welcomed by researchers in the field. The manuscript would benefit from a minor revision.

It is very unfortunate that the initial developer of this method, Kotaro Kelley, passed away. I am sure if he had not, his research in the field would have brought even more breakthrough discoveries and methodological developments. My position as a reviewer does not allow me to comment on this sad

event in the context of the manuscript suitability for publication, but as a person and scientist, I would be grateful to the editorial board for taking it into account.

My comments:

Some parts of the introduction sound somewhat superficial in the current version.

1. Lines 22–24: “Cryo-FIB/SEM has emerged from within the field of cryo-EM as the method for obtaining the highest resolution structural information of complex biological samples in-situ in native and non-native environments.”

Cryo-FIB/SEM cannot be mentioned in this context independently from cryo-electron tomography, because cryo-FIB/SEM itself (cryo-ASV) does not result in “the highest resolution”. Cryo-FIB/SEM is the method of sample preparation for high-resolution cryo-ET or microcrystalline electron diffraction (MicroED). It is worth considering a modification of this sentence.

2. Lines 42–44: “Conventional cellular cryo-focused ion beam milling scanning electron microscopy (cryo-IB/SEM)¹ is developing as a fruitful method for structural studies of cells thinner than about 10 μm .”

The same comment as above applies. It is not obvious what you mean by the “conventional cryo-FIB/SEM”. Please define. I expect this is plunge freezing of cells attached to the EM grid, followed by cryo-FIB/SEM milling?

3. Lines 47–49: “However, the broad application of conventional cryo-FIB/SEM with reasonable throughput is often challenged by specimens with dimensions smaller than about 2 μm and which adopt a preferred orientation on the grid.”

Is the preferential orientation the only existing limitation of the “conventional cryo-FIB/SEM”? In my opinion, a very limited vitrification depth of plunge-freezing in one of the most important ones.

4. Lines 50–63: “Three previous approaches have been used to address issues of specimens thicker than about 50 μm prior to FIB-milling lamellae for imaging in a transmission electron microscope (TEM)...”

This is difficult to understand, especially after you were talking about samples that are smaller than 2 microns in your previous sentence. Can you specify what issues you mean? Why exactly 50 microns? Can you better explain how cryo-ASV is used for the lamellae milling?

High-pressure freezing (HPF) was used not only for cryo-ASV, but for many other studies mentioned in the citation 12 in this paragraph. In my opinion, HPF is the key point. One can easily make a thick sample that avoid the preferential orientation, but such a sample cannot be vitrified unless one uses HPF.

5. Lines 64–75: The reader will appreciate a short explanation. Why is it named the “Waffle method”?

6. Lines 76–79: I would mention that 100 micron long polar tube is folded inside 2(?) micron spores, otherwise it gives impression that you milled at least 100 micron thick samples.

7. Lines 88–89: “to different cryo-FIB/SEM hardware, namely the Aquilos 2 combined with automated lamellae milling.”

Only a single piece of hardware is mentioned. Please add others or modify this sentence.

The Results section requires reorganization to avoid self-repetition.

8. Line 115: Please add an explanation (or a citation) for the connection between the carrier surface roughness and lamellae curtaining. It is not self-obvious for readers outside of FIB-SEM field.

9. Please reorganize the chapters “Overall workflow” and “Prepare hardware and waffling the sample parts” to remove duplications.

10. Consider conversion of the chapter “Hardware requirements and recommendations” into a supplementary text/table to improve the text flow.

11. Line 137: “Current is reduced and the lamellae is milled down to about 1.5 μm thick while tilting the stage $\pm 1^\circ$ with a tab left beside the notch.”

Can you provide reasons for this tilting? It is not self-obvious.

12. Line 142: “The finished 14 lamellae are then sputtered with several nanometers of conductive platinum.”

Do you mean a cold deposition using gas injection system inside the chamber of cryo FIB-SEM of a

sputter coating in argon atmosphere outside of the FIB-SEM?

13. Line 252: Can you provide any initial interpretation of the bumps seen in polar tubes?

14. Would it make sense to reorganize the parts of Figure 3 in two vertical columns "Conventional" and "Waffle" to unambiguously follow the explanation?

15. The chapter linked to Figure 3 speaks exclusively about the orientation problem solution, but it does not mention the throughput and concentration improvement mentioned in the Figure 3 title. Milling large and deep trenches and lamellae in Waffles should require a much longer time comparing to the conventional thin samples on grids. Can you comment on the throughput improvement more quantitatively (cell sections in lamellae/milling day)?

16. Lines 276–277: Can you explain your reasons for milling an extremely large lamella that is too thick and thus useless for high-resolution cryo-ET? Structure overview for correlative imaging?

Any comments on what we can see inside representative tomograms?

17. Lines 285–290: Can you avoid self-repetitive sentence wording?

Discussion

18. Lines 295–296: See comment 1.

19. Lines 299–302: I strongly recommend that the vitrification issue is placed first (see comment 4).

20. Line 309: "Additionally, the Waffle Method may present opportunities for cell-cell interaction studies."

This sentence needs further explanations, as it is not shown in results.

21. Line 310: "Several developments and implementations were key to the success of the development of the Waffle Method" can be removed to avoid repeating the same phrases in the following sentences.

22. Lines 365–367: The stresses within the Waffle squares can represent the major limitation of the technique. It would be useful to explain potential reasons, such as differences in thermal shrinkage between the grid metal and the vitreous sample. Can you propose potential solutions?

23. Line 452: "Shorter planchette hats may be used to optimize waffle thickness, and possibly development of grids with shorter grid bars"

It is not clear to me how shorter planchette hats (do you mean thinner?) can reduce waffle thickness. Is the Waffle setup compatible with other high-pressure freezers? If yes, it is worthwhile mentioning it to avoid a commercial bias towards the Wohlwend HPF apparatus.

Methods

24. Please check the order or numbering of the tables.

Figure legends

25. Line 832: "multiple" is not relevant here. Please identify the specific problems.

26. Line 847: "dotted" do you mean dashed?

27. Figure 4: provide thickness of the tomographic slices.

28. Figure 4: I am sure that readers will appreciate short descriptions of structural details visible in every representative image and the corresponding videos.

29. Line 860: please specify the "inhibitive issues".

30. What are the molecules in Figure 4b?

Reviewer #3:

Remarks to the Author:

The manuscript from Kelley et al describes a new approach to improve sample preparation for samples that are going to be processed through cryo-FIB milling. Sample preparation in this field is not yet a bottleneck, but the proposed approach helps to increase the quality and the throughput of this process.

Furthermore, the authors present a demonstration that the Waffle preparation method could allow an improved single-particle sample preparation.

The article is extremely well written and the protocol is described in great detail. I find that the article

as it stands is definitely worth publishing, but for a journal of the calliper of Nature Communications, I find that there are a couple of points that should be addressed. I suspect these will be easily addressed by the authors.

In the lab, we replicated the proposed method following the pre-print available for freezing cells in and we found that there is a failure rate associated with the carriers not opening after the HPF run. This appears to be mildly improved by a careful polishing of the carrier surface following the described protocol. It would be useful to have a more detailed description of this matter, in particular a statistic around the failure rate and what was done to improve it.

The method described for the stress relief in this article is superb, although it is slightly more complex to fabricate compared to the other published method, this offers a number of advantages especially in terms of general stability during transfer. We found it to be more stable.

Despite the core of the article focuses on cell and tissue preparation for in situ structural studies, I find that this method has the greatest potential in solving the preferred orientation and the particle segregation at the air-water interface. As the authors described, this method indeed appears to help in these regards. In order to validate that the method provides an actual benefit, I would expect a structural determination and a comparison with the number of particles required through regular preparation. Here, I do not intend to say that the structural determination through single-particle analysis might be possible (I see no reason why this would not be the case), but the absence of the holey carbon/gold film and the polished lamella surface tend to charge more. This will influence the amount of beam-induced movement. Furthermore, the minimum thickness of a lamella is greater than what is achievable with other grid preparation methods (blotting or blot free) therefore again increasing the effect of beam-induced movement and decreasing the signal to noise. A comparison of the number of particles that are required to achieve a certain resolution and an evaluation of the number of particles that need to be discarded would most likely answer my query.

Personally, I find that this comparison would add a significant value to the article, highlighting probably the strongest advantage of this method as this would solve a major bottleneck in cryo-EM.

Response to Reviewers (Kelley, et al.)

We would like to thank the reviewers for their analysis and comments, which have significantly improved our manuscript. We provide individual responses to reviewers' comments and questions in the following pages. My apologies for the delay. Frankly, returning to this manuscript is difficult for me due to Kotaro's death and the surrounding circumstances. -AJN

Point-by-point responses (original comments in black; responses in blue; changes in the manuscript are highlighted in blue)

Reviewer #1:

The manuscript summarizes a new approach to make a wider range of samples amenable for focused ion beam milling and cryo electron tomography, without the need for extra lift-out hardware. The Waffle method described by the authors has several advantages to other methods, such as: samples of tens of microns in thickness can be vitrified and processed, and there is no issue with preferred orientation of certain samples on the support grid. The authors provide detailed instructions on how to use the method and provide multiple convincing examples. With even more automation coming to the field, this method can soon become even more accessible and streamlined. The Waffle method described here is very elegant and is of wide interest to the community.

We thank the reviewer for their overall favorable evaluation.

A few minor comments:

- Could the authors please, provide one more figure summarizing all the main steps from having the waffled sample in the FIB-SEM and until the polished lamellae? A perfect figure (in my view) would have a column with schematic drawings (with average dimensions of lamellae/cuts/notches and their geometries), a column with actual example images, and some text next to them to describe each step. This would be a great aid for the future user of the method. Potentially the information from table 1/2 can be added to such a figure.

We have added the requested figure as Supplemental Figure 3.

- Could the authors add typical dimensions of the stress-relief notch that they used to the Supp. Figure 7? The length, height, width, in order for readers to reproduce this closely. Please, also add which currents were used for notch milling to the Table 1.

We have added the requested information to the main text, new Supplemental Figure 3, Supplemental Figure 7, and the Methods. For a 200 mesh grid with a ~20 micron thick waffle, we mill 200 nm wide notch forms for ~2 minutes at 0.3 nA. The length, height, and milling time depend on the slab size, at what thickness the notch is milled, and user preferences. We have included our current design in new Supplemental Figure 3 from the question above.

Reviewer #2:

This manuscript by Kelley et al. presents a new workflow, named the Waffle method, for cryo-electron tomography (cryo-ET) sample preparation. The Waffle method is based on the use of high-pressure freezing applied to samples attached to, or suspensions deposited on, standard electron microscopy grids. After freezing, these grids, transferred into a cryo-scanning microscope equipped with a focused ion beam (FIB) and thin vitreous lamellae, are prepared under cryogenic conditions. Tomographic data are then recorded from these lamellae using a transmission cryo-electron microscope.

The Waffle method combines important findings that enable researchers to overcome several significant limitations of the existing cryo-lamellae preparation approaches. The use of high-pressure freezing allows the Waffle method to be applied to samples that cannot be reliably vitrified by plunge freezing, which is commonly used to fabricate cryo-lamellae from flat cells growing on, or attached to, EM grids. Thus, the approach proposed by the authors can be applied to a wide range of samples, such as cells, tissues, and embryos. On the other hand, the authors have demonstrated the potential of the method to solve orientation and denaturation problems related to the water–air interface in single-particle reconstruction methods. Importantly, the Waffle method makes it possible to prepare the lamellae directly on the sample grid, which helps to achieve higher throughput and does not require expensive equipment, such as the lift-out method, for samples a few tens of microns thick.

Additionally, the authors have proposed an arrow-shaped notch geometry, which is an interesting addition to existing developments for mechanical lamella stabilization.

To summarize, the methodological developments presented in this manuscript are of great importance for the further development of in situ structural analysis approaches and will be enthusiastically welcomed by researchers in the field. The manuscript would benefit from a minor revision.

We thank the reviewer for their overall favorable evaluation.

It is very unfortunate that the initial developer of this method, Kotaro Kelley, passed away. I am sure if he had not, his research in the field would have brought even more breakthrough discoveries and methodological developments. My position as a reviewer does not allow me to comment on this sad event in the context of the manuscript suitability for publication, but as a person and scientist, I would be grateful to the editorial board for taking it into account.

We agree. Kotaro was endlessly enthusiastic and creative. Thank you for your kind words.

My comments:

Some parts of the introduction sound somewhat superficial in the current version.

1. Lines 22–24: “Cryo-FIB/SEM has emerged from within the field of cryo-EM as the method for obtaining the highest resolution structural information of complex biological samples in-situ in native and non-native environments.”

Cryo-FIB/SEM cannot be mentioned in this context independently from cryo-electron tomography, because cryo-FIB/SEM itself (cryo-ASV) does not result in “the highest resolution”. Cryo-FIB/SEM is the method of sample preparation for high-resolution cryo-ET or microcrystalline electron diffraction (MicroED). It is worth considering a modification of this sentence.

We have clarified by introducing and using the following combined acronym throughout the manuscript where appropriate: cryo-FIB/SEM-ET.

2. Lines 42–44: “Conventional cellular cryo-focused ion beam milling scanning electron microscopy (cryo-IB/SEM)¹ is developing as a fruitful method for structural studies of cells thinner than about 10 μm .”

The same comment as above applies. It is not obvious what you mean by the “conventional cryo-FIB/SEM”. Please define. I expect this is plunge freezing of cells attached to the EM grid, followed by cryo-FIB/SEM milling?

We have added our definition of the term “conventional cryo-FIB/SEM” to the first sentence in the Introduction, which has the meaning that the reviewer inferred.

3. Lines 47–49: “However, the broad application of conventional cryo-FIB/SEM with reasonable throughput is often challenged by specimens with dimensions smaller than about 2 μm and which adopt a preferred orientation on the grid.”

Is the preferential orientation the only existing limitation of the “conventional cryo-FIB/SEM”? In my opinion, a very limited vitrification depth of plunge-freezing in one of the most important ones.

We agree and have added vitrification to this motivating sentence.

4. Lines 50–63: “Three previous approaches have been used to address issues of specimens thicker than about 50 μm prior to FIB-milling lamellae for imaging in a transmission electron microscope (TEM)...”

This is difficult to understand, especially after you were talking about samples that are smaller than 2 microns in your previous sentence. Can you specify what issues you mean? Why exactly 50 microns?

We have added a reference for the 50 micron limit (Schaffer et al., 2019). The typical depth of focus for a gallium FIB beam in a cryo-FIB/SEM is ~50 microns.

Can you better explain how cryo-ASV is used for the lamellae milling?

High-pressure freezing (HPF) was used not only for cryo-ASV, but for many other studies mentioned in the citation 12 in this paragraph. In my opinion, HPF is the key point. One can easily make a thick sample that avoid the preferential orientation, but such a sample cannot be vitrified unless one uses HPF.

We know of instances outside of the literature where researchers are using cryo-ASV (rather, semi-automated) to help localize areas of interest when milling chunks of HPFed sample, and one instance of cryo-FLM being used to localize within said chunks. We have added these potential usages as possible ways to improve the Waffle Method to the Discussion.

5. Lines 64–75: The reader will appreciate a short explanation. Why is it named the “Waffle method”?

We have added an explanation to the end of the Introduction shortly after the term ‘waffle’ is introduced. Historically, it was a name that quickly caught on during development because an HPFed sample on a grid in an SEM with grid bars drawn on looks like a waffle. The name also helps explain the method by simply showing a picture of a breakfast waffle with syrup filling in the squares (ie. sample filling grid bars on an EM grid). It was a cute name that stuck.

6. Lines 76–79: I would mention that 100 micron long polar tube is folded inside 2(?) micron spores, otherwise it gives impression that you milled at least 100 micron thick samples.

We have made this clarification in the text by changing the word ‘possess’ to ‘enclose’.

7. Lines 88–89: “to different cryo-FIB/SEM hardware, namely the Aquilos 2 combined with automated lamellae milling.”

Only a single piece of hardware is mentioned. Please add others or modify this sentence.

Since the previous version of the manuscript, we have successfully installed and used SerialFIB on our FEI Helios cryo-FIB/SEM to perform rough and fine milling of waffle lamellae. We have added this to the Introduction and Discussion.

Additionally, we have also heard from numerous researchers in the field of successful application of Waffle Method on separate Leica ICE HPFs, TFS Aquiloses, and a Zeiss Crossbeam cryo-FIB/SEM. We have added these to the Discussion.

The Results section requires reorganization to avoid self-repetition.

8. Line 115: Please add an explanation (or a citation) for the connection between the carrier surface roughness and lamellae curtaining. It is not self-obvious for readers outside of FIB-SEM field.

We substantiate the need for a sufficiently smooth waffle surface in Supplemental Figure 4, which is referenced and explained after the Overall workflow so as not to clutter the protocol. We have added ‘as we substantiate herein’ to this line as a self-citation.

9. Please reorganize the chapters “Overall workflow” and “Prepare hardware and waffling the sample parts” to remove duplications.

We separated these sections purposefully. The ‘Overall workflow’ section is meant to be generic and agnostic to specific hardware. The ‘Prepare hardware and waffling the sample’ subsection specifically describes the preparation of the *E. hellem* spores in the parent section. We have added a clarification at the outset of the Results section to clarify the purpose of the ‘Overall workflow’ section.

10. Consider conversion of the chapter “Hardware requirements and recommendations” into a supplementary text/table to improve the text flow.

We have converted this section to a Supplementary Table. We agree that it improves the text flow.

11. Line 137: “Current is reduced and the lamellae is milled down to about 1.5 μm thick while tilting the stage $\pm 1^\circ$ with a tab left beside the notch.”

Can you provide reasons for this tilting? It is not self-obvious.

We sometimes find that tilting during finer milling helps keep the lamella more uniformly thick. Milling deep lamellae (ie. in the direction of the FIB beam) has a tendency to produce lamellae with varying thicknesses due to FIB focusing variability. We have added an explanation to the text.

12. Line 142: “The finished 14lamellae are then sputtered with several nanometers of conductive platinum.”

Do you mean a cold deposition using gas injection system inside the chamber of cryo FIB-SEM or a sputter coating in argon atmosphere outside of the FIB-SEM?

We are referring to the latter. This has been clarified in the text.

13. Line 252: Can you provide any initial interpretation of the bumps seen in polar tubes?

The data presented here for the polar tube are from test samples with only a few tomograms and we are currently in the process of collecting more data and analyzing these using both segmentation and sub-tomogram averaging. Currently there is not enough data and the analysis is not far enough along to include unambiguous interpretation, and for this reason we are hesitant to include anything beyond sample preparation-related information in the current manuscript. Our hope is that this will form an important part of a forthcoming study on the polar tube that will provide many more insights.

14. Would it make sense to reorganize the parts of Figure 3 in two vertical columns “Conventional” and “Waffle” to unambiguously follow the explanation?

We prefer that Fig 3 e,f be split into two columns as it is currently to show clearly that preferred orientation is solved. Also, (b) and (c) are not exactly analogous because one is after milling and one is before, and we don't have the corresponding post-milling

overhead image of all of the E. Hellum lamellae together. So, in our opinion the horizontal separation of 'Conventional' and 'Waffle' and vertical separation of preferred orientations in e,f are optimal for this figure.

15. The chapter linked to Figure 3 speaks exclusively about the orientation problem solution, but it does not mention the throughput and concentration improvement mentioned in the Figure 3 title.

We have added a mention on throughput and concentration in the main text corresponding to Figure 3.

Milling large and deep trenches and lamellae in Waffles should require a much longer time comparing to the conventional thin samples on grids. Can you comment on the throughput improvement more quantitatively (cell sections in lamellae/milling day)?

We have added a rough comparison of the throughput improvement to the Discussion.

16. Lines 276–277: Can you explain your reasons for milling an extremely large lamella that is too thick and thus useless for high-resolution cryo-ET? Structure overview for correlative imaging?

While the lamella may be too thick for high-resolution analysis, it may still be useful for medium-resolution analysis. The main purpose for milling this lamella, however, is to demonstrate feasibility (specifically: survivability) of lamellae much larger than the already large lamellae in the manuscript. Milling such very large lamellae uniformly down to 200 nm or thinner would be the next step and would allow for even higher throughput milling and greater amounts of contextual information. As this is a developing method, we are providing as much information as possible to the reader so that they know the limits and the potential of the method for their own research. We have clarified the purpose for this lamella in the Discussion.

Any comments on what we can see inside representative tomograms?

We have added annotations to some known features in the cell tomogram videos.

17. Lines 285–290: Can you avoid self-repetitive sentence wording?

We have combined these sentences to avoid repetition.

Discussion

18. Lines 295–296: See comment 1.

Updated.

19. Lines 299–302: I strongly recommend that the vitrification issue is placed first (see comment 4).

We agree and have changed the ordering throughout the text.

To be completely clear, the Waffle Method was first successfully applied to the E. Hellum sample, which has concentration/throughput and preferred orientation issues, but not vitrification issues. Thus the first two issues were initially highlighted. Now that the method has been around for a year, we see both internally and through comments from researchers elsewhere that vitrification is a main benefit of the Waffle Method's way of creating lamellae on the grid.

20. Line 309: "Additionally, the Waffle Method may present opportunities for cell–cell interaction studies."

This sentence needs further explanations, as it is not shown in results.

We have added a sentence in the Results section for Figure 4b highlighting that interactions between neighboring cells, as the figure depicts, may be studied.

21. Line 310: "Several developments and implementations were key to the success of the development of the Waffle Method" can be removed to avoid repeating the same phrases in the following sentences.

We have removed the sentence.

22. Lines 365–367: The stresses within the Waffle squares can represent the major limitation of the technique. It would be useful to explain potential reasons, such as differences in thermal shrinkage between the grid metal and the vitreous sample. Can you propose potential solutions?

In practice since our manuscript was submitted, we have found that notch milling absorbs the vast majority of stresses that may be imparted on the lamellae, regardless of

the amount of sample vitrified onto the grid. We no longer see this as an issue and have thus removed this bullet point. Thank you for bringing this to our attention.

23. Line 452: “Shorter planchette hats may be used to optimize waffle thickness, and possibly development of grids with shorter grid bars”

It is not clear to me how shorter planchette hats (do you mean thinner?) can reduce waffle thickness.

The planchette hats we use usually have a flat side and a cupped side (ie. a side with a flat indentation). By ‘shorter’ we are referring to a shorter indentation, or no indentation at all (ie. flat). Now that the reviewer has brought up this point, however, we see that this sentence is not consistent with the rest of the manuscript where the flat side of the planchette hats are presumed to be the grid-facing sides. Thus we have removed the first part of this sentence.

Is the Waffle setup compatible with other high-pressure freezers? If yes, it is worthwhile mentioning it to avoid a commercial bias towards the Wohlwend HPF apparatus.

We have personally tested waffling on a Leica ICE with success, and we have heard of several other labs successfully using their Leica ICE HPF to make waffles. This has been added to the Discussion.

Methods

24. Please check the order or numbering of the tables.

Updated.

Figure legends

25. Line 832: “multiple” is not relevant here. Please identify the specific problems.

We have updated the Figure legend.

26. Line 847: “dotted” do you mean dashed?

Updated.

27. Figure 4: provide thickness of the tomographic slices.

Thickness added.

28. Figure 4: I am sure that readers will appreciate short descriptions of structural details visible in every representative image and the corresponding videos.

We have added annotations to some known features in Figure 4 and the tomogram videos.

29. Line 860: please specify the “inhibitive issues”.

Some potential issues are specified in the main text when referencing this figure. We have updated the Figure legend to include the most common issues.

30. What are the molecules in Figure 4b?

Assuming that the reviewer is asking about the molecules in Figure 5b instead of 4b, this is an internal T20S proteasome single particle reconstruction. We have clarified this in the Figure legend.

Reviewer #3:

The manuscript from Kelley et al describes a new approach to improve sample preparation for samples that are going to be processed through cryo-FIB milling. Sample preparation in this field is not yet a bottleneck, but the proposed approach helps to increase the quality and the throughput of this process.

Furthermore, the authors present a demonstration that the Waffle preparation method could allow an improved single-particle sample preparation.

The article is extremely well written and the protocol is described in great detail. I find that the article as it stands is definitely worth publishing, but for a journal of the calliper of Nature Communications, I find that there are a couple of points that should be addressed. I suspect these will be easily addressed by the authors.

We thank the reviewer for their overall favorable evaluation.

In the lab, we replicated the proposed method following the pre-print available for freezing cells in and we found that there is a failure rate associated with the carriers not opening after the HPF run. This appears to be mildly improved by a careful polishing of the carrier surface following the described protocol. It would be useful to have a more detailed description of this matter, in particular a statistic around the failure rate and what was done to improve it.

We currently have a 100% success rate with planchette hat-grid separation for waffle HPF grids. We currently follow the workflow described in the paper, plus two addendums: 1) We use solid brass planchette hats. We would caution against using metal-coated planchette hats because the abrasion from the sanding/polishing step may induce corrosion of the metals over time. 2) When applying 1-hexadecane to the planchette hats, we now place the planchette hats grid-facing side up on filter paper wetted with 1-hexadecane and then place a drop of 1-hexadecane on the grid-facing side of each hat and let it sit for 15 minutes before immediately assembling the waffle for high-pressure freezing. We have updated the text with this information and have added an additional image as part (f) of Supplemental Figure 1 to show the 1-hexadecane drops incubating on the grid-facing sides of four planchette hats.

The method described for the stress relief in this article is superb, although it is slightly more complex to fabricate compared to the other published method, this offers a number of advantages especially in terms of general stability during transfer. We found it to be more stable.

We thank the reviewer for experimentally evaluating notch milling with their equipment.

Despite the core of the article focuses on cell and tissue preparation for in situ structural studies, I find that this method has the greatest potential in solving the preferred orientation and the particle segregation at the air-water interface. As the authors described, this method indeed appears to help in these regards. In order to validate that the method provides an actual benefit, I would expect a structural determination and a comparison with the number of particles required through regular preparation. Here, I do not intend to say that the structural determination through single-particle analysis might be possible (I see no reason why this would not be the case), but the absence of the holey carbon/gold film and the polished lamella surface tend to charge more. This will influence the amount of beam-induced movement. Furthermore, the minimum thickness of a lamella is greater than what is achievable with other grid preparation methods (blotting or blot free) therefore again increasing the effect of beam-induced movement and decreasing the signal to noise. A comparison of the number of particles that are required to achieve a certain resolution and an evaluation of the number of particles that need to be discarded would most likely answer my query.

Personally, I find that this comparison would add a significant value to the article, highlighting probably the strongest advantage of this method as this would solve a major bottleneck in cryo-EM.

We're glad that the reviewer appreciates the potential of applying the Waffle Method to single particle samples. We agree that a full study into the feasibility of this potential solution to air-water interface problems would be of interest to the entire field of cryo-EM. A full study would require optimization for several single particle samples, milling thickness optimization, full single particle cryo-EM/ET post-processing with and without waffling, and high-resolution analysis of potential denaturation regions for each sample. However, in this manuscript we included this preliminary single particle waffle lamella only in the Discussion to highlight one of several potential uses for the Waffle Method that have not been fully explored. This manuscript is primarily intended to introduce the Waffle Method in application to cellular specimens. We recognize that it may be used as a platform for further development in several different directions. Thus we feel that a full study in this particular direction is well outside of the scope of this manuscript.

Reviewers' Comments:

Reviewer #1:

Remarks to the Author:

The authors have fully addressed my questions. I recommend this paper for publication

Reviewer #2:

Remarks to the Author:

I thank the authors for the changes and corrections.

I recommend this manuscript for publication.

Response to Reviewers (Kelley, et al.)

We would like to thank the reviewers for their analysis. We provide individual responses to reviewers' comments below.

Point-by-point responses (original comments in black; responses in blue)

Reviewer #1:

The authors have fully addressed my questions. I recommend this paper for publication

We thank the reviewer for their overall favorable evaluation.

Reviewer #2:

I thank the authors for the changes and corrections.
I recommend this manuscript for publication.

We thank the reviewer for their overall favorable evaluation.